# Alluvial cover controlling the width, slope and sinuosity of bedrock channels

Jens Martin Turowski[1]

[1] Helmholtzzentrum Potsdam, German Research Centre for Geosciences GFZ, Telegrafenberg, 14473 Potsdam, Germany

*Correspondence to*: Jens M. Turowski (turowski@gfz-potsdam.de)

**Abstract.** Bedrock channel slope and width are important parameters for setting bedload transport capacity and in stream-profile inversion to obtain tectonics information. Channel width and slope development are closely related to the problem of bedrock channel sinuosity. It is therefore likely that observations on bedrock channel meandering yields insights into the development of channel width and slope. Active meandering occurs when the bedrock channel walls are eroded, which also

drives channel widening. Further, for a given drop in elevation, the more sinuous a channel is, the lower is its channel bed slope in comparison to a straight channel. It can thus be expected that studies of bedrock channel meandering give insights into width and slope adjustment and vice versa. The mechanisms by which bedrock channels actively meander have been debated since the beginning of modern geomorphic research in the 19th century, but a final consensus has not been reached. It has long been argued that whether a bedrock channel meanders actively or not is determined by the availability of sediment

relative to transport capacity, a notion that has also been demonstrated in laboratory experiments. Here, this idea is taken up by postulating that the rate of change of both width and sinuosity over time is dependent on bed cover only. Based on the physics of erosion by bedload impacts, a scaling argument is developed to link bedrock channel width, slope and sinuosity to sediment supply, discharge and erodibility. This simple model built on sediment-flux driven bedrock erosion yields the observed scaling relationships of channel width and slope with discharge and erosion rate. Further, it explains why sinuosity

evolves to a steady state value and predicts the observed relations between sinuosity, erodibility and storm frequency, as has been observed for meandering bedrock rivers on Pacific Arc islands.

## 1 Introduction

Bedrock channels are the conveyer belts of mountain regions. Once sediment produced on hillslopes by mass wasting reaches a channel, it is evacuated along the stream network. In the process, the moving particles act as tools for bedrock erosion and

the river adjust until it reaches a steady state. Then, channel morphology, parameterised for example by the width and bed slope of the channel, stays constant over time, and the vertical erosion rate adjusts to match tectonic uplift. In turn, width and slope determine the transport capacity of the channel and thus the channel's efficiency in evacuating sediment. As a results, channel long profiles and width can be used as indicators for local tectonic process rate, and uplift rates and histories can in principle be calculated from morphologic characteristics of the channel network (e.g., Kirby and Whipple, 2001; Roberts and

White, 2010; Wobus et al., 2006b). Conventionally, the inversion of channel morphology to obtain tectonic information has focussed on long profiles or slope, despite the observation that channel width also adjusts to tectonic forcing (e.g., Duvall et al., 2004; Lavé and Avouac, 2001; Yanites et al., 2010). For reliable inversion we thus need a model that can predict the effects of uplift both on channel width and slope (e.g., Turowski et al., 2009), and possibly other morphologic parameters.

Davis (1893) sparked a long-standing debate in geomorphology when he described the meanders of the Osage River as inherited from a prior alluvial state of the channel. Although this explanation is still frequently encountered to explain why bedrock channels are sinuous, even Davis' contemporaries argued that active meandering occurs in incised channels (e.g., Winslow, 1893). By now, numerous field observations of features such as cut off meander loops and gentle slip-off slopes in
inner meander bends have confirmed that actively meandering bedrock channels exist and are common (e.g., Barbour, 2008; Ikeda et al., 1981; Mahard, 1942; Moore, 1926; Seminara, 2006; Tinkler, 1971). However, the mechanics of bedrock river meandering are still debated and have recently attracted research interest (e.g., Johnson and Finnegan, 2015; Limaye and Lamb, 2014), since the meandering problem is closely related to the problems of terrace formation, lateral planation, gorge eradication, and bedrock channel width (c.f., Cook et al., 2014; Finnegan and Balco, 2013; Turowski et al., 2008a). In fact,
active meandering is dependent on lateral erosion of the channel walls and is therefore directly related to the adjustment of channel width. Similarly, meandering lengthens the channel over a given drop of height and thereby reduces bed slope. Thus, it seems likely that observations on bedrock channel sinuosity are informative also for the study of channel width and slope, and vice versa. While the power-law scaling of channel width and slope with discharge with typical exponents of ~1/2, positive for width and negative for slope, is widely acknowledged (e.g., Lague, 2014; Snyder et al., 2003; Whipple, 2004; Whitbread
et al., 2015; Wohl and David, 2008), observations of the scaling relationships of sinuosity are less commonly discussed. In a detailed study of Japan, Stark et al. (2010) demonstrated that lithology poses a first order control on the sinuosity of actively incising bedrock channels, with weak sedimentary rocks displaying higher values of a regional measure of sinuosity than volcanic or crystalline lithologies. Once this influence was accounted for, a positive trend of sinuosity with the variability of precipitation emerged, quantified by typhoon-strike frequency or by the fraction of days with rainfall exceeding a threshold.
This positive trend with storm frequency could generally be confirmed for other islands of the Pacific Arc, including Taiwan, Borneo, New Guinea, and the Philippines (Stark et al., 2010). The prediction of the relationships observed by Stark et al. (2010) remains a benchmark for any theory of bedrock channel meandering, but an explanation is lacking so far. Further, in addition to observations on channel bed slope and width, the sinuosity scaling provides another line of evidence for validation of general models of bedrock channel morphology.

Sinuosity increases when, within a channel bend, the bank at the outer bend erodes faster than at the inner bend. Alluvial meander theory relates this imbalance in lateral erosion to hydraulics (e.g., Edwards and Smith, 2002; Einstein, 1926; Ikeda et al., 1981). Within the bend, there are higher flow speeds in the outer bend than in the inner bend. Erosion rate and therefore the meander migration rate is assumed to be dependent on the velocity difference. In contrast, in many bedrock channels,

erosion is driven by particle impacts in the two most common fluvial erosion processes plucking and impact erosion (e.g., Beer et al., 2017; Chatanantavet and Parker, 2009; Cook et al., 2013; Sklar and Dietrich, 2004). Abrasion means the erosion due to impacts of moving bedload particles. Plucking means the removal of larger blocks of rock by hydraulic forces. In the latter process, impacts drive crack propagation and thus the production of pluckable blocks, which is also known as macro-abrasion

(Chatanantavet and Parker, 2009). In environments where particle impacts drive erosion, the outer bends of meanders are particularly prone to erosion as particle trajectories detach from flow lines and can thus impact the walls (e.g., Cook et al., 2014). If bedrock channel sinuosity is indicative of past climate, as Stark et al. (2010) suggested, then bedrock channels need the ability to first adjust to the required sinuosity and second to keep this sinuosity constant over long time periods, while continuing vertical incision. The latter feat can be achieved either by stopping lateral erosion once the required sinuosity is

reached or by maintaining a balance of those processes that increase sinuosity and those that decrease it. The only known mechanism for decreasing sinuosity is meander cut-off. However, cut-off can only occur if the channel meanders actively, and it is only effective when sinuosity is high. The sinuosity of bedrock channels observed by Stark et al. (2010) span a wide range of values, including low sinuosities that cannot be kept steady with recurring cut off. Thus, it seems unlikely that the cut-off mechanism can balance lateral erosion rates at low sinuosity to achieve a steady state. The argument suggests that channels

cease or at least strongly decrease active meandering once they have reached the steady state sinuosity, but why they do this is an open problem. This raises the question as to when and why some bedrock channels actively meander while others do not. In general, two lines of argument have been proposed to answer this question.

The first line of argument asserts that the process of bedrock erosion controls lateral erosion rates, and local lithology

determines this process and thus whether a channel actively meanders or not. Johnson and Finnegan (2015) compared two bedrock channels in the Santa Cruz Mountains, California, USA, one actively meandering in a mudstone sequence, the other one incising without meanders into a sandstone. While both lithologies showed similar strength when dry, the mudstone lost strength through slaking due to wetting-drying cycles and could thereafter be eroded by clear water flows. In this case, essentially, active meandering could be achieved by a similar hydraulic mechanism as has been described for alluvial streams

(e.g., Edwards and Smith, 2002; Ikeda et al., 1981; Seminara, 2006). Moore (1926) likewise described an influence of lithology on the meanders of streams on the Colorado Plateau – there, meanders can be found in sandstone units, while in weaker shales, the valleys are wide and straight. However, Moore (1926) did not describe different erosion mechanisms (e.g., slaking, impact erosion) for the two lithologies, and it is unclear what causes the different channel behaviour in his study region. While the slaking mechanism should be more efficient in a variable climate due to more frequent wetting-drying cycles, in line with

Stark et al.'s (2010) observations, it fails to explain why a stream can continue incising while maintaining a constant sinuosity. Further, Stark et al. (2010) described sinuous bedrock channels in a range of lithologies, including hard crystalline rock, where slaking erosion is likely not important.

The second line of argument builds on the relative availability of sediment in the channel. In resistant bedrock, erosion is driven by the impacts of moving particles in the two most common fluvial bedrock erosion processes, abrasion and plucking. The increasing erosion rate with increasing relative sediment supply is known as the tools effect (e.g., Cook et al., 2013, Sklar and Dietrich, 2004). Conversely, stationary sediment residing on the bed can protect the bedrock from impacts. This is known

as the cover effect (e.g., Sklar and Dietrich, 2004; Turowski et al., 2007), which has been argued to play a key role in the partitioning of vertical to lateral erosion (e.g., Hancock and Anderson, 2002; Turowski et al., 2008a). Moore (1926) suggested that whether a bedrock river actively meanders or not depends on the relative availability of sediment, a notion that was later investigated experimentally by Shepherd (1972). In Shepherd's (1972) experiments, a sinuous channel was cut into artificial bedrock made of sand, kaolinite and silt, which was not erodible by clear water flow. Base level, water discharge and sediment

supply were kept constant over the entire run time of 73 hours. At first, all sediment could be entrained by the flow and the channel cut downwards, without changing the planform pattern. But as the channel bed slope declined due to erosion over the course of the experiment, patches of sediment formed on the inside bends and the channel started to widen and to meander actively. Shepherd (1972) suggested that lateral erosion rates stayed similar throughout the entire run, while vertical erosion rates declined due to the increasing importance of the cover effect. Thus, at first, lateral and vertical erosion were balanced

such that channel width kept constant over time, while the later decrease in vertical incision led to channel widening and ultimately migration and active meandering.

Shepherd's (1972) experimental observations point to fundamental importance of bed cover in setting bedrock channel width and meandering dynamics. In this paper, I develop a physics-based scaling argument to explain the observed scaling of bedrock

channel width, slope, and sinuosity. The argument is motivated by the behaviour of the experimental channel of Shepherd (1972) and is built on the fundamental assumption that bed cover controls lateral erosion. It exploits general considerations and observations about bedload transport, and process knowledge of fluvial bedrock erosion. Since channel morphology is set by the partitioning of erosion between bed and banks, the problem is approached by assessing under which conditions lateral erosion can occur and how these conditions relate to channel bed cover. The physical considerations lead to a model of incising

channels with stable width, slope and sinuosity. Model predictions are compared to observed scaling relationships of bedrock channel width and slope with discharge, drainage area and erosion rate, and to the sinuosity scaling observed by Stark et al. (2010).

**2 Model development**

Previous attempts of predicting bedrock channel morphology can be grouped in four classes (Shobe et al., 2017, also gave a

recent overview). (i) 1D-models using a shear stress or stream power formulation (e.g., Seidl et al., 1994; Whipple, 2004). These models capture the fundamental scaling of slope with discharge, and, to an extent, of slope with erosion rate, but need to make assumptions on width-discharge scaling for closure (see Lague, 2014, for a review). Zhang et al. (2015) described a

morpho-dynamic model that also captures alluvial dynamics and includes both tools and cover effects. However, this model is restricted to channels with macro-rough beds, i.e., topography with a relief that is a substantially larger than the dominant grain size. (ii) 1D-models that treat channel width explicitly, but, instead of assuming a width-discharge scaling, make an alternative assumption to close the system of equations. Suggested assumption have been a constant width-to-depth ratio (Finnegan et al., 2005) or optimization of energy expenditure (Turowski et al., 2007). These models have been proposed assuming a shear stress or stream power erosion law (Finnegan et al., 2005; Turowski et al., 2009), as well as sediment-flux-dependent erosion laws including either just the cover effect (Yanites and Tucker, 2010) or both tools and cover effects (Turowski et al., 2007). For the shear stress erosion model, the closing assumption has at least been partially validated against models treating the cross-sectional evolution of a channel (Turowski et al., 2009). Although these models can predict a range of observed scaling relations, especially if sediment flux effects are included in the erosion model (see Turowski et al., 2007; Yanites and Tucker, 2010), they suffer from a lack of physics-based arguments for connecting lateral erosion to channel morphology and from the essential arbitrariness of the closing assumption. (iii) 2D-models that explicitly model some aspects of the width dynamics. For a shear stress erosion law, Stark (2006) used a slanted trapezoidal channel shape, while Wobus et al. (2006a) and Turowski et al. (2009) described models with fully adjustable channel cross section. Lague (2010) used a trapezoidal cross section and included the cover effect in his formulation. The success of these models in predicting scaling relationships is similar to the models of class (ii), but none of the models published so far includes all aspects of the current understanding of the process physics of fluvial bedrock erosion. Further, none of these models properly treats fully alluviated beds, where alluvial channel processes dominate, which can strongly affect long-term erosional dynamics and channel adjustment time scales (cf. Turowski et al., 2013). (iv) 3D-models that, to some extent, resolve the interaction of hydraulics and sediment transport and their effect on bedrock erosion (e.g., Inoue et al., 2016; Nelson and Seminara, 2011, 2012). These models are generally numerically expensive and have not been used to investigate scaling relations on the reach to catchment scale.

As a results, we lack a model that is rooted in the current understanding of process physics and can predict channel width, slope, and sinuosity on the catchment scale. Here, inspired by the experiments described by Shepherd (1972), I put forward the fundamental postulate that the partitioning between lateral and vertical erosion, and therefore width adjustment and sinuosity development, is controlled by a single variable, bed cover. Parameters such as sediment supply, river sediment transport capacity and bed topography directly control cover, but they only indirectly control the distribution of erosion by altering bed cover. Formalizing the observations made in Shepherd's (1972) experiments, we can make the following statements: (i) At low degrees of cover, channel width stays constant and the channel does not meander actively, and (ii) channel widening and active meandering commences when a threshold cover is exceeded. In section 2.1, based on considerations based on the physics of erosion by particle impacts and of bedload transport, I develop a scaling argument for bedrock channel width. In section 2.2, the slope of the channel is discussed. In section 2.3, the argument is applied to the development of bedrock channel sinuosity.

## 2.1 Lateral erosion and bedrock channel width

Consider a straight bedrock channel with sub-vertical walls. The general direction of water and particle discharge is parallel to the walls, although we can expect some lateral motion due to secondary currents and turbulent fluctuations. As bedrock erosion is achieved by particle impacts, the requirement for lateral erosion is a sideward deflection of travelling particles such that they (i) reach and impact the wall, and (ii) upon impact, have enough energy to cause damage to the rock. Lateral motion of sediment particles can be driven by secondary currents, turbulent fluctuation and momentum diffusion (e.g., Diplas et al., 2008; Parker, 1978), cross-stream diffusion of particle paths (Seizilles et al., 2014), gravitationally-driven migration on cross-sloping beds (e.g., Parker et al., 2003), or by sideward deflection by obstacles on the bed (Beer et al., 2017; Fuller et al., 2016). For given conditions – hydraulics, bed morphology, sediment supply, and grain characteristics – we can define a sideward deflection length scale $d_{xy}$ for every point on the bed, which depicts the maximum distance a particle can be deflected sideward while still causing erosion. This length scale should be a function of hydraulics or transport capacity, channel bed slope, channel curvature, bed roughness, sediment properties (size, shape, density), and possibly of the erodibility of the bedrock via the threshold for erosion. Crucially, it can be expected that $d_{xy}$ can vary considerably over short distances both along and across the channel, depending on bed topography and the local distribution of roughness and alluvial cover. For the construction of a reach-scale model of bedrock channel morphology, we need to first find the relevant point within each cross section and the corresponding $d_{xy}$ that determines lateral erosion (which we call $d_x$), and then the relevant cross section and the corresponding $d_x$ that determines channel width in the reach (which we call $d$). For a given channel, the propensity to lateral erosion then depends on the ratio of the sideward deflection length scale $d$ and the channel width $W$ (Fig. 1). In a channel with a width much larger than $d$, only bedload moving close to the walls, precisely, within a distance $d$ of the walls, can contribute to lateral erosion. In contrast, in a channel with $W \sim d$, all bedload can contribute to lateral erosion.

In general, a bedrock channel widens only when bedload particles impact the walls, i.e., in the framework proposed above, that some bedload is moving within a distance $d$ from the walls. For purpose of illustration, consider a narrow, straight bedrock channel with $W \sim d$ (Fig. 1). Due to frequent particle impacts on the walls, lateral erosion rates are high and the channel widens. This leads to a decrease in the areal sediment concentration and thus a decrease in the number of bedload particles that can cause lateral erosion. At some point bedload impacts on the wall become so unlikely that widening ceases. The channel has reached a steady state width. However, this argument does not capture the entire story, since we have neglected vertical incision. Next, this aspect will be included in the consideration and the ratio $d/W$ will be tied to one of the common observables in bedrock channel morphology, the covered fraction of the bed $C$ (e.g., Sklar and Dietrich, 2004; Turowski and Hodge, 2017).

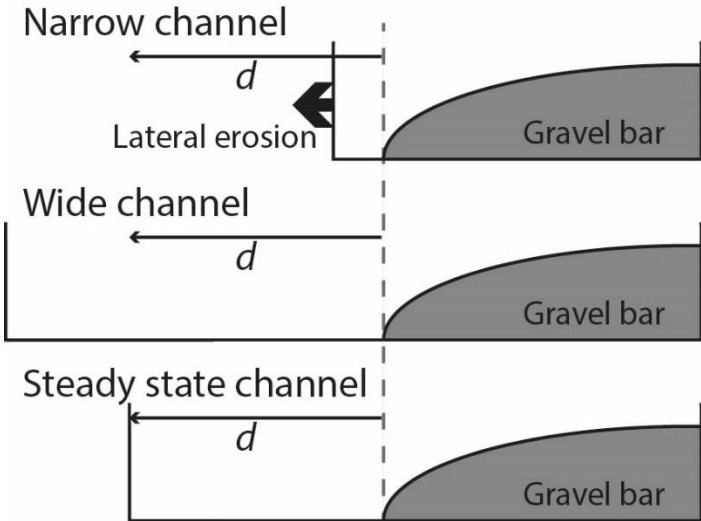

**Figure 1: Illustration of how the sideward deflection length scale *d* and the channel width interact to determine lateral erosion. The dashed vertical line shows the relevant deflection point within the cross section. Top: in a narrow channel, particles that are laterally deflected a distance *d* may hit the wall and cause erosion. The channel widens. Center: in a wide channel, the deflected particles do**
**not reach the wall. No lateral erosion occurs. Conversely, few particles travel over the bedrock bed near to the wall. Sufficient tools to drive the vertical erosion of the bed are only available within the distance *d* of the deflection point. An inner channel with the steady state width is formed. Bottom: in a steady state channel, the channel walls are just out of reach of the deflected particles.**

The relative efficiency of lateral to vertical erosion has been tied to bed cover in conceptual-theoretical arguments (e.g.,
Hancock and Anderson, 2002; Moore, 1926), experimental observations (e.g., Finnegan et al., 2007; Johnson and Whipple, 2010; Shepherd, 1972) and field studies (e.g., Beer et al., 2016; Johnson et al., 2010; Turowski et al., 2008a). Using a combination of experiments and modelling, it has been argued that the fraction of covered bed area is an adequate proxy for the reduction of erosion due to the shielding effect of sediment on the reach scale (Turowski and Bloem, 2016). Consequently, cover *C* is commonly defined as the covered bed area fraction, i.e., the bed area covered by sediment $A_{cover}$ divided by the total
bed area of the considered reach $A_{tot}$. Normalising by the length of the considered reach *L*, we can write *C* also as a ratio between two length scales, the relevant covered width $W_{cover}$ (which could be a reach average or the covered width for the cross section relevant for setting lateral erosion rates) and the channel width *W*.

$$C = \frac{A_{cover}}{A_{tot}} = \frac{A_{cover}/L}{A_{tot}/L} = \frac{W_{cover}}{W}$$

(1)
At low sediment supply, cover is low to non-existent. Sufficient tools for incision are available only where the particle stream concentrates. There, an inner channel is formed, and so the channel narrows (e.g., Finnegan et al., 2007; Johnson and Whipple, 2010). To a similar effect, in wide channels, several longitudinal grooves tend to form at low sediment supply (Inoue et al.,

2016; Wohl and Ikeda, 1997). One of these draws most sediment and water and, after some time, develops into an inner channel that captures the entire water and sediment supply. At high sediment supply, the bed is covered by sediment, which reduces vertical erosion to zero. Lateral erosion occurs in a strip just above the cover, where bedrock is exposed and tools are abundant (Beer et al., 2016; Turowski et al., 2008a). The channel widens. We can formalise the observations outlined above by relating

the rate of change of channel width, $dW/dt$, to relative sediment supply $Q_s^*$, which is the ratio of bedload supply $Q_s$ to transport capacity $Q_t$ (Fig. 2). At $Q_s^* = 0$, lateral erosion and therefore $dW/dt$ is also zero, due to the lack of erosive tools. For small $Q_s^*$, the channel narrows and $dW/dt$ must be negative. For high $Q_s^*$, the channel widens and $dW/dt$ must be positive. Since cover $C$ is generally related to $Q_s^*$ (e.g., Sklar and Dietrich, 2004; Turowski and Hodge, 2017; Turowski et al., 2007), a similar relationship must arise between $dW/dt$ and cover. At a critical value, $Q_c^*$ or $C_c$, the channel behaviour switches from narrowing

to widening and $dW/dt = 0$. This is the only point where the channel both has a steady width and incises vertically with a finite erosion rate. At the critical cover, the distance of bedload particles from the walls needs to be equal to the sideward deflection length scale $d$. If $d$ is larger than this typical distance, frequent impacts will occur on the channel walls and the channel widens (Fig. 1). If it is smaller, few bedload particles move in the vicinity of the walls, leading to a lack of erosive tools, and the bed near the walls is not eroded. An inner channel forms for which the above condition is true.

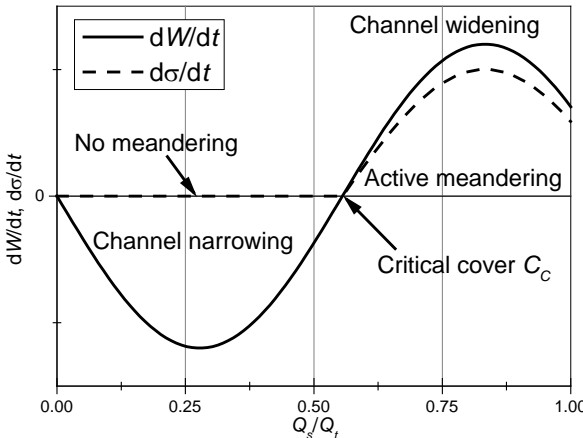

**Figure 2: Schematic relation between the rate of change of width $dW/dt$ (black line) and sinuosity $d\sigma/dt$ (dashed line) with relative sediment supply $Q_s/Q_t$. At low supply, no sediment particles impact the walls, the channel narrows, and does not meander actively. At high supply, frequent sediment impacts on the channel walls drive lateral erosion, leading to channel widening and active**
**meandering. At the critical cover, the rate of change of width is zero. The exact position of this point depends on absolute channel width.**

As can be seen from the following argument, the critical cover $C_c$ must depend on channel width and should indeed scale with $d/W$. Chatanantavet and Parker (2008) demonstrated with experiments that in wide straight channels in the cover-dominated domain, alternating gravel bars formed. Inoue et al. (2016) modelled this situation and found that a meandering thread of alluvial material between alternating submerged gravel bars migrates downstream over uniformly eroding bedrock, leading to

a channel with a symmetric bedrock cross section. From studies on alluvial rivers it is known that the main path of bedload particles in a straight channel with submerged bars is offset from the main path of water and the thalweg (e.g., Bunte et al., 2006; Dietrich and Smith, 1984; Julien and Anthony, 2002). Gravel bedload moves across the bar, enters the thalweg at the bar centre, traverses it and climbs the next downstream bar at its head (Fig. 3). Similarly, it has been observed that in a partially alluviated bedrock channel, sediment moves from patch to patch or from bar to bar (Ferguson et al., 2017; Hodge et al., 2011).

However, the precise bedload path over partially covered bedrock has not yet been described. For the following argument, I make two main assumptions: (i) the bedload path determined by Bunte et al. (2006) for gravel bed channels with alternating submerged bars applies also to bedrock channels (Fig. 3). This assumption is plausible and is adopted since there is a lack of direct relevant data. (ii) The sideward deflection length of bedload is largest at the edge of alluvial patches or bars in the direction of the uncovered bedrock (Fig. 4). This assumption is made for three reasons. First, the bedrock is typically smoother

than the alluviated section and provides less impediment to particle movement, in particular to sideward deflection toward the uncovered part of the cross section (cf. Chatanantavet and Parker, 2008; Ferguson et al., 2017; Hodge et al., 2011, 2016). Second, at the edge of bars, the alluvium provides roughness elements that can lead to sideward deflection (cf. Beer et al., 2017, Fuller et al., 2016). Third, at this point the velocity vector of the bedload particles has a large cross-stream component; in fact, it is at its maximum (Fig. 3). In a channel with steady state width, bedload particles at this point should just fail to reach

the wall, and we can assume that the sideward deflection length scale $d$ is approximately equal to the uncovered width (Fig. 3). At steady state, we therefore expect that the following relation holds:

$$C_c = \frac{W_c}{W} = \frac{W - d}{W} = 1 - \frac{d}{W}$$

(2)

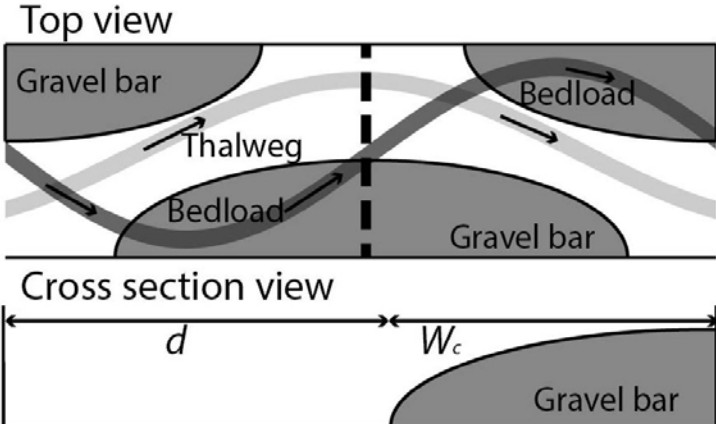

Figure 3: Top: Schematic drawing of the top view of the channel with alternating gravel bars (dark grey), thalweg and main water pathway (light grey), and main bedload path way (transparent dark grey) after Bunte et al. (2006). Uncovered bedrock is depicted in white. Bottom: Cross section across the centre of a bar (dotted black line in the top view), where the bedload path crosses from the bar into the uncovered channel. This cross section is relevant for setting the reach-scale channel width, since the sideward deflection of bedload particles toward the left-hand wall should be maximised (cf. Fig. 4). At steady state, the uncovered width within the cross section should be equal to the sideward deflection length scale $d$, and the relation $d + W_c = W$ should hold (cf. Fig. 1; eq. 2).

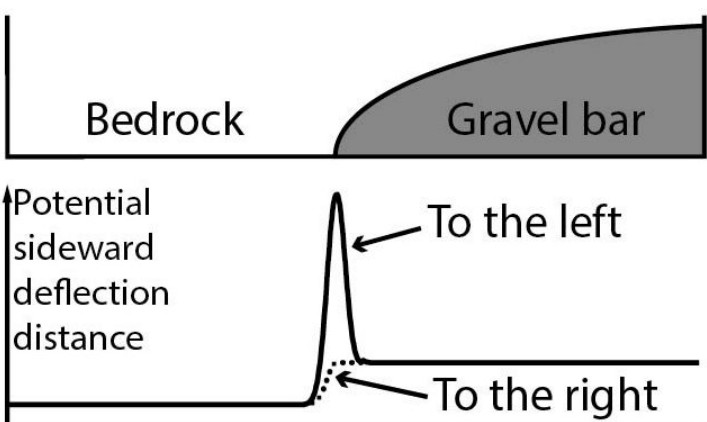

Figure 4: The potential sideward deflection distance is larger over alluvium than bedrock, since roughness elements facilitate sideward deflection of moving particles. However, the same roughness elements block path of the deflected particles, thus limiting the total distance. The largest deflection distances occur at the boundary between alluvium and bedrock towards the bedrock bed. Only where the particle stream intersects this point can large sideward deflection distances be achieved.

Using the equation for critical cover (eq. 2), we can relate channel width to vertical erosion rate using one of the established models for incision (e.g., Auel et al., 2017; Sklar and Dietrich, 2004). I assume a sediment-flux dependent erosion law, including tools and cover effect, of the form

$$E = k\frac{Q_s}{W}(1 - C)$$

5    (3)

Here, $E$ is the vertical erosion rate, and $k$ is a parameter that describes the erodibility of the rock. As before, $Q_s$ is the upstream bedload sediment supply. Note that in the original saltation-abrasion model (Sklar and Dietrich, 2004), $k$ depends explicitly on hydraulics, but consistently, in all of the field and laboratory studies where all relevant parameters have been measured, this dependency has not been found (Auel et al., 2017; Beer and Turowski, 2015; Chatanantavet and Parker, 2009; Inoue et al., 10    2014; Johnson and Whipple, 2010). At steady state, $C = C_c$. Substituting eq. (2) into eq. (3) and solving for width, we obtain an equation for the steady state width of bedrock channels.

$$W = \sqrt{\frac{kQ_s d}{E}}$$

(4)

15    **2.2 Channel bed slope**

To extend the argument to channel bed slope, an additional equation is needed relating bed cover to sediment supply and transport capacity. Several equations have been suggested in the literature, including the linear decline model (Sklar and Dietrich, 2004) and the negative exponential (Turowski et al., 2007). Recently, Turowski and Hodge (2017) derived a model 20    of the form

$$C = \left(1 - e^{-\frac{Q_s}{M_0 UW}}\right)\frac{Q_s}{Q_t}$$

(5)

Here, $e$ is the base of the natural logarithm, $U$ is the average bedload particle speed, and $M_0$ is the minimum mass per bed area necessary to completely cover the bed, which is dependent on grain size (Turowski, 2009, Turowski and Hodge, 2017). Note 25    that eq. (5) reduces to the linear decline model at high sediment supply, i.e., for large $Q_s$.

We can write the bedload transport capacity per unit width as a power function of both discharge $Q$ and channel bed slope $S$ (e.g., Rickenmann, 2001; Smith and Bretherton, 1972)

$$\frac{Q_t}{W} = K_{bl}Q^m S^n$$

30    (6)

Here, $K_{bl}$ is a constant and it has been argued that the exponents $m$ and $n$ typically take values between 1 and 4 (Barry et al., 2004; Smith, 1974). Note that in eq. (6), the threshold of motion of bedload has been neglected. Such a threshold is generally accepted to be relevant for bedload motion (e.g., Buffington and Montgomery, 1997) and will become important when linking sinuosity to storm frequency. Assuming steady state at the critical cover $C_c$, substituting eqs. (2) and (6) into (5) and solving for $S$, we get

$$S = \left(1 - e^{-\frac{Q_s}{M_0 U W}}\right)^{1/n} \left(\frac{Q_s}{K_{bl}(W - d)}\right)^{1/n} Q^{-\frac{m}{n}}$$

(7)

## 2.3 Sinuosity

At a given location, lateral erosion and therefore the development of curvature and sinuosity is of course dependent on local conditions such as the channel width, bed slope and long-stream curvature (e.g., Cook et al., 2014; Howard and Knutson, 1984; Inoue et al., 2016). But rather than trying to predict the detailed evolution of the planform pattern, here I propose a reach-scale view of sinuosity development. As is conventional, sinuosity $\sigma$ is defined as the ratio of the total channel length $L_C$ to the straight length $L_V$ from end to end. Note that this is equivalent to the ratio of valley slope $S_V$ to channel slope $S$.

$$\sigma = \frac{L_C}{L_V} = \frac{S_V}{S}$$

(8)

Sinuosity can only increase if the walls of the channel are eroded. Thus, the rate of change of sinuosity $d\sigma/dt$ should be zero when $dW/dt$ is negative. Sinuosity development commences at the same critical cover $C_c$ that marks the transition from channel narrowing to widening and $d\sigma/dt$ should be positive when $dW/dt$ is positive also (Fig. 2). However, we need to slightly adjust the picture that has been advanced in section 2.1, since instead of a straight channel, we are now dealing with a curved channel. Further, channel curvature is varying along the stream. As before, lateral erosion should stop once the channel walls are outside of the reach of particle impacts. Due to curvature, particle trajectories detach from water flow lines and wall erosion rates can be expected to be highest in regions with the highest curvature (e.g., Cook et al., 2014; Howard and Knutson, 1984). Point bars develop in the inside bends, providing roughness for sideward deflection (Fig. 5). Substantial particle impacts can thus be expected in the outside bends, probably a little downstream of the bend apex (cf. Fig. 5). The rest of the argument can stay essentially the same: lateral erosion stops once the bedrock wall is just outside of the reach of the deflected particles. The bedrock channel is driven to a steady state at which $C = C_c$. At this point, sinuosity development ceases and the channel essentially stalls itself in its active meandering. Treating valley slope as an independent parameter, eq. (8) can be substituted into eq. (7) and solved for sinuosity to obtain

$$\sigma = \left(1 - e^{-\frac{Q_s}{M_0 U W}}\right)^{-1/n} \left(\frac{K_{bl}(W - d)}{Q_s}\right)^{1/n} S_V Q^{\frac{m}{n}}$$

(9)

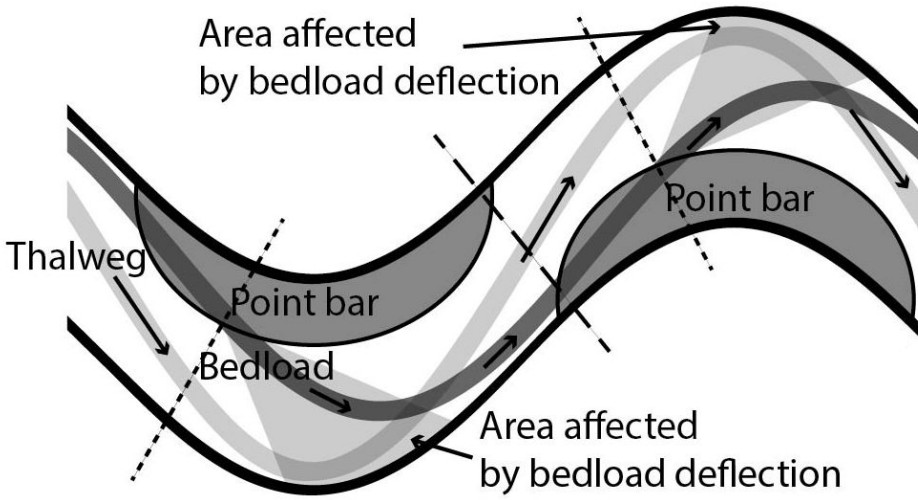

**Figure 5: Schematic illustration of the thalweg (light grey) and gravel bedload path (dark grey) through a meandering channel, after the observations of Dietrich and Smith (1984) and Julien and Anthony (2002). Uncovered bedrock is depicted in white and gravel point bars in dark grey. Flow is from left to right. Dotted lines show the relevant cross section for particle deflection. Areas that are presumably affected by bedload particle deflection and should lead to wall erosion are shaded in light grey. The dashed line is placed at the inflection point of the channel centre line.**

## 3. Comparison to observations

In this section, I will compare the model to field and laboratory observations. First, I will interpret the experiments of Shepherd (1972) in light of the arguments that lead to the model equations. Then, I will compare field observations to the predictions by the equations. Since for most field sites many essential parameters are not known, I will focus on accepted scaling relations. Lague (2014) has summarised the available data for geometry and dynamics of bedrock channels and has identified six lines of evidence that any model needs to match. Two of these are related to transient channel dynamics and knickpoint migration. Since the model developed in the present paper is only concerned with steady state channels, the remaining lines of evidence, namely slope-area scaling, slope-erosion rate scaling, width-area scaling, and width-erosion rate scaling, are discussed below. To these I add the two scaling relations for the sinuosity of channels, sinuosity-erodibility scaling, and sinuosity storm-frequency scaling, as observed by Stark et al. (2010).

For the comparison with field data, I use six data sets that include information on erosion rates, with scaling relationships as compiled by Lague (2014) (Table 1). Two of these data sets arise from studies of rivers crossing a fault, the Bakeya, Nepal (Lavé and Avouac, 2001) and the Peikang river, Taiwan (Yanites et al., 2010). The data for the Bagmati, Nepal (Lavé and

Avouac, 2001), was not used, since a tributary joins the stream within the studied reached, supplying unknown amounts of both water and sediment and thereby altering boundary conditions (see Lague, 2014; Lavé and Avouac, 2001; Turowski et al., 2009). Four of the data sets arise from studies comparing different catchments that are thought to be in a topographic steady state along a gradient in uplift rate with otherwise comparable conditions. These are channels from the Siwalik Hills, Nepal (Kirby and Whipple, 2001; re-analysed by Wobus et al., 2006b), the Mendocino Triple Junction (Snyder et al., 2000), Eastern Tibet (Ouimet et al., 2009), and the San Gabriel Mountains (DiBiase et al., 2010). I did not use the data from the Santa Inez Mountains (Duvall et al., 2004), since a lack of coarse bedload in these mudstone channels has been reported (Whipple et al., 2013). There, impact erosion may not be the dominant erosion process, which could alter channel processes, morphology and dynamics. The channels studied by Tomkin et al. (2003) and Whittaker et al. (2007), draining catchments with strong long-stream gradients in uplift rate, are under-constrained for the purpose of model comparison, since the variation of erosion rates and therefore sediment supply along the stream is unknown.

Table 1: Data sets and scaling exponents used for model evaluation, as reported by Lague (2014).

| | | River / Region (observations) Domain (predictions) | Scaling exponents | | Reference |
|---|---|---|---|---|---|
| | | | Width-erosion rate | Slope-erosion rate | |
| Channels crossing a fault | Observations | Bakeya, Nepal | -0.63 | 0.49 | Lavé and Avouac, 2001 |
| | | Peikang, Taiwan | -0.42 | 0 | Yanites et al., 2010 |
| | Model prediction | Tools-dominated | -0.5 | 0.12-0.47 | |
| | | Cover-dominated | -0.5 | 0.07-0.33 | |
| Steady state catchments | Observations | Eastern Tibet | N.A. | 0.65 | Ouimet et al., 2009 |
| | | San Gabriel Mountains | 0 | 0.49 | DiBiase et al., 2010 |
| | | Mendocino Triple Junction | 0 | 0.25 | Snyder et al., 2000 |
| | | Siwalik hills | N.A. | 0.93 | Kirby and Whipple, 2001; Wobus et al., 2006b |
| | Model prediction | Tools-dominated | 0 | 0.27-1.05 | |
| | | Cover-dominated | 0 | 0.14-0.67 | |

For parts of the discussion it is useful to work with two approximations for the cover equation, eq. (5), both for the sake of algebraic simplicity and ease of argument. First, in the tools-dominated domain, cover is scarce and bedrock erosion rate is controlled by the availability of tools. Then, $Q_s/W$ is small and the exponential term can be approximated with a first-order Taylor expansion, reducing eq. (5) to

$$C_{tools} = \frac{Q_s{}^2}{M_0 U W Q_t}$$

(10)

Then, we can reduce the first term in the slope and the sinuosity equations, eqs. (8) and (10), which yields

$$\left(1 - e^{-\frac{Q_s}{M_0 UW}}\right) \approx \frac{Q_s}{M_0 UW}$$

(11)

Second, in the cover-dominated domain, tools are abundant, but most of the bed is covered. Then, the erosion rate is set by the fraction of the exposed bedrock. Sediment supply per unit width $Q_s/W$ is large, the exponential term vanishes, and we retrieve

the linear model (Sklar and Dietrich, 2004)

$$C_{cover} = \frac{Q_s}{Q_t}$$

(12)

Then, the first term in the slope and the sinuosity equations, eqs. (8) and (10), reduces to one

$$\left(1 - e^{-\frac{Q_s}{M_0 UW}}\right) \approx 1$$

10    (13)

The cover-dominated approximation (eqs. 12 and 13) is likely most relevant for the data discussed here. It is known that many actively incising bedrock rivers exhibit substantial cover at least at low flow (Meshkova et al., 2012; Tinkler and Wohl, 1998; Turowski et al., 2008b; 2013), and it seems likely that for many rivers the sideward deflection length scale $d$ is much smaller than the channel width (formally, $W>>d$, leading to $W - d \approx W$), implying substantial cover at steady state. Therefore, it can

be expected that the tools-dominated approximation (eqs. 10 and 11) is only relevant for small headwater streams or for channels that do not receive much coarse sediment, for example due to an upstream reservoir.

### 3.1 Shepherd's (1972) experiment

Shepherd's observations have been described in detail in the introduction. From a model perspective, consider a stream that

re-incises its bed after a base level drop. At constant sediment supply, as the stream incises, bed slope and therefore transport capacity decreases. As a result, cover increases (eq. 5). At some point the critical cover $C_c$ is exceeded and the stream starts active meandering. Meandering lengthens the flow path and therefore also decreases bed slope and transport capacity. The subsequent increase in cover leads, at some point, to full cover stopping vertical incision. Once the steady state width is reached, lateral erosion drops to zero. Then, the stream also stops active meandering. It essentially stalls itself and reaches a

steady state for sinuosity. The described scenario is equivalent to the one observed by Shepherd (1972), although the stalling phase was not reached in his experiments.

### 3.2 Channel width

A number of studies report the sensitivity of channel width to uplift rate (for summaries of the available data, see Lague, 2014;

Turowski et al., 2009; Whipple, 2004; Yanites and Tucker, 2010). Several different behaviours have been observed (see also Table 1). In comparisons of channels in catchments that differ only by uplift rate, channel width was comparable at similar

drainage areas, indicating that there was no response to uplift rate (Snyder et al., 2003; DiBiase and Whipple, 2011). In another study, Duvall et al. (2004) found narrower channels in catchments with higher uplift rates, but this could also be related to the lack of coarse bedload in the mudstone channels (Whipple et al., 2013). Similarly, some channels display a typical width-area scaling despite strong gradients in uplift rate (Tomkin et al., 2003; Whittaker et al., 2007). In contrast, channels crossing an
uplifting fault block tend to narrow (Lavé and Avouac, 2001; Yanites et al., 2010).

According to the proposed model, steady state channel width scales with the square root of the product of sediment supply $Q_s$, erodibility $k$, and sideward deflection length scale $d$, and inversely with the square root of the vertical incision rate $E$ (eq. 4). The different response of channel width in studies comparing different channels in areas with gradients in uplift rate (no
channel narrowing) and those that looked at single channels crossing an uplifting fault block (channel narrowing) can be explained by the role of sediment flux. I will discuss the latter case first.

When a channel crosses from a region without uplift into an uplifting fault block, water discharge and sediment load stay the same, provided there are no tributaries or major hillslopes sediment sources. Thus, in the width equation (eq. 4), sediment
supply $Q_s$ is constant and the channel responds by increasing erosion rate $E$ to match the increased uplift rate. Provided that $k$ and $d$ are independent of erosion rate, the channel narrows and channel width should scale with incision rate to the power of -1/2. Two of the cases mentioned above allow a direct evaluation of this prediction. In the Bakeya River (Lavé and Avouac, 2001), the width-erosion rate scaling exponent is -0.63, and in the Peikang River (Yanites et al., 2010), the scaling exponent is -0.42 (Table 1), both close to the predicted value of -1/2.

In catchments in a topographic steady state, the channel geometry adjusts such that the long-term incision rate matches the long-term uplift rate or base level lowering rate. Averaged over the catchment, the sediment supply can be written in terms of erosion rate $E$ and catchment area $A$.

$$Q_s = \beta E A$$

25   (14)
Here, $\beta$ is the fraction of material that contributes to bedrock erosion, i.e., the bedload fraction. The steady state channel width equation (4) then becomes

$$W = \sqrt{k\beta dA}$$

(15)
As vertical incision rate $E$ cancels out, steady state channel width in this case is independent of uplift rate. This is in agreement with field observations (Table 1). Equation (15) also predicts the typical scaling of channel width $W$ with the square root of drainage area $A$. However, it is likely that both the gravel bedload fraction $\beta$ and the sideward deflection length scale $d$ vary in a systematic fashion with drainage area. The bedload fraction tends to decrease with increasing drainage area (e.g., Turowski et al., 2010), possibly even to the extent that bedload supply $Q_s$ is independent of drainage area (see Dingle et al., 2017). There

are additional complications that arise from non-linear averaging of sediment supply both with varying floods and stochastically varying bedload supply. Further, the bedload fraction $\beta$ is likely dependent on erosion rate $E$, in a currently unknown way. At the moment little is known about how $d$ varies along a stream. I will return to this point in the discussion.

**3.3 Channel bed slope**

A power law scaling of slope with drainage area with an exponent of -1/2 is widely assumed to be indicative of steady state bedrock channels.

$$S = k_s A^{-\theta}$$

(16)

This relationship is known as Flint's law (Flint, 1974), although it has earlier been studied by Hack (1957). The pre-factor $k_s$ is called the steepness index and the exponent $\theta$ is called the concavity index. For the concavity index, a range of values of 0.4-0.6 is often reported (Lague, 2014). Whipple (2004) gives a range of 0.4-0.7 for actively incising bedrock channels in homogenous substrates with uniform uplift, while higher concavities (0.7-1.0) are associated with decreasing uplift rates in the downstream direction. Using data from catchments where erosion rate have been constrained using cosmogenic nuclides, Harel

et al. (2016) found a median value of the concavity index of 0.52±0.14, with a similar range as reported by Whipple (2004). It seems, therefore, that the observed variability in the value of the concavity index is higher than is generally acknowledged, with observed values as low as 0.4 and as high as 1. In comparisons of channels in steady state landscapes, the steepness index $k_s$ has been observed to increase with incision rate according to a power law, with an exponent ranging from 0.25 to 0.93 (Table 1), derived from four data sets (Lague, 2014). The two channels crossing a fault block exhibit different scaling. The Bakeya

(Lavé and Avouac, 2001) shows a positive relationship with an exponent of about 0.49, while for the Peikang (Yanites et al., 2010), little to no slope changes in response to uplift have been reported.

The brief summary of observations above implies that a model should be able to account for the following observations. (i) Slope should decrease with drainage area according to a power law with an exponent value varying between about 0.4 and

0.7. (ii) The exponent may be altered if there are gradients in uplift rate along the stream; in particular, a downstream decrease in uplift may drive the concavity index up to higher values of up to about 1. (iii) In channels draining catchments in a topographic steady state, the steepness index should increase with uplift rate according to a power law with an exponent value varying between about 0.25 and 1.0. (iv) In channels crossing a fault block, slope may or may not increase in response to uplift.

Often, the concavity index in the slope-area relationship is related to a slope-discharge scaling by assuming that discharge scales with drainage area following a relationship of the form

$$Q = k_h A^c$$

(17)

Here, $k_h$ and $c$ are catchment-specific values describing the hydrology. In particular, the exponent $c$ takes a value of 1 if the exchange of water with ground water storage and evapotranspiration are spatially uniform in the catchment (e.g., Snyder et al., 2003). For natural data, the value of $c$ is dependent on the discharge chosen for the regression. For the long-term mean annual discharge, various effects should average out and $c$ should be close to 1 (Dunne and Leopold, 1978, as cited by Snyder et al., 2003). Leopold et al. (1964) reported values between 0.70 and 0.75 for bankfull discharge. When transforming the observed values of the concavity index of the slope-area scaling to an exponent of the slope-discharge relationship, we thus obtain a range of values for the slope-discharge exponent of 0.4-1.0 for steady state channels in uniform conditions and 0.7-1.4 for channels with a downstream decrease in uplift rate.

In the model equation (eq. 7), slope scales with discharge to a power of $-m/n$. Many bedload transport equations can be written in the form of equation (6) (Smith and Bretherton, 1972), and the theoretical values of $m$ and $n$ depend on the chosen equation. For example, the Einstein (1950) bedload equation yields $m = n = 2$ (Smith and Bretherton, 1972), while Meyer-Peter and Müller (1948) type equations yield $m = 1$ and $n = 1.5$ (Rickenmann, 2001). However, in the latter case, the linear scaling arises only if the threshold of bedload motion is neglected and is thus valid only for large floods. Rickenmann (2001) argued that $n = 2$ gives a better fit for both laboratory and field data at stream gradients larger than 3%. However, he also included relative roughness as a separate predictor, which is implicitly dependent on slope. If written out explicitly, the dependence on slope should be stronger, with values of $n$ potentially much larger than 2 (see also Nitsche et al., 2011; Schneider et al., 2015). Measured $m$-values are usually much larger than those derived from models. For example, Bunte et al. (2008) reported $m$-values ranging from about 7.5 to 16, using data obtained during the snow-melt period of American streams with portable bedload traps. Analysing bedload data sampled with Helley-Smith pressure difference samplers from a large number of streams, Barry et al. (2004) found values of $m$ in the range of about 1.5-4.0. They used drainage area instead of slope in their transport equation, and the data given in their paper do not allow a re-evaluation in terms of slope. Nevertheless, a regression of channel bed slope of the sites against drainage area yields an exponent of -0.48, giving an estimate of $n \approx 7.1$. From the mentioned cases, it is clear that depending on the choice of equation or data set, a wide range of values for the $m$ and $n$ scaling exponents can be obtained. Finally, it needs to be noted that most bedload data and bedload transport equations in the literature have been derived for channels with a mobile bed. Bedload equations specifically for natural bedrock channels are not known to the author. In addition to the explicit relationship of slope and discharge, slope is implicitly related to discharge via sediment supply, channel width and the sideward deflection length scale, all of which could depend on discharge or drainage area.

Out of the discussed approaches, the field data evaluation by Rickenmann (2001) may be most appropriate for the purpose at hand, since the data were derived from long term-monitoring of deposition in retention basins. The time scale of the data is thus closer to the time scales of bedrock erosion and channel adjustment than the near-instantaneous measurements used for example by Barry et al. (2004). This would yield values of $m = 1$ and $n = 2$, and a ratio $m/n = 0.5$ (Rickenmann, 2001). For the

remainder of the discussion, I will use this case as standard, as well as a range of $n$-value of 1.5-7 for evaluating possible ranges of the values of scaling exponents.

The equations and the discussion are considerably simplified in the tools- or cover-dominated approximations (see eqs. 10-13). In the tools-dominated case, channel bed slope is given by

$$S_{tools} = \left( \frac{Q_s{}^2}{M_0 U K_{bl} W(W-d)} \right)^{1/n} Q^{-\frac{m}{n}}$$

(18)

Here, we can recognise two different cases. First, consider narrow headwater channels. There, the sideward deflection length scale $d$ is of the order of the channel width $W$. As a result, slope depends strongly on the actual values of $d$ and $W$ and their scaling with other morphological parameters, e.g., bed roughness. I will not further consider this case, as there are few relevant data available. Second, consider a wide channel carrying little coarse sediment, for instance due to an upstream reservoir. Then, $W \gg d$ and eq. (18) reduces to

$$S_{tools} = \left( \frac{Q_s{}^2}{M_0 U K_{bl} W^2} \right)^{1/n} Q^{-\frac{m}{n}}$$

(19)

Since bedload particle speed $U$ is dependent on hydraulics, there is an implicit dependence of $U$ on slope and discharge, which needs to be taken into account. With standard assumptions on flow velocity and shear stress (Appendix A), eq. (19) becomes

$$S_{tools} = k_{tools} \left( \frac{E}{kd} \right)^{\frac{3+\alpha}{4n+\alpha+1}} (Q_s)^{\frac{5-\alpha}{4n+\alpha+1}} (Q)^{-\frac{4m-2\alpha+2}{4n+\alpha+1}}$$

(20)

Here, $k_{tools}$ is assumed to be constant (see eq. A9, Appendix A), and $\alpha$ is a constant that typically takes a value of 0.6 (e.g., Nitsche et al., 2011). In the case of a steady state channel crossing an uplifting fault block, $Q_s$ and $Q$ can be considered constant and only $E$ varies. In this case, the discharge exponent is equal to -0.5 as long as $m/n = 1/2$. For $n = 1.5$, the dependence on erosion rate and erodibility yields an exponent of 0.47, with decreasing values as $n$ increases (it evaluates to 0.375 for $n = 2$, 0.20 for $n = 4$ and 0.12 for $n = 7$). For a channel in a steady-state landscape, we can substitute eq. (14) to obtain

$$S_{tools} = k_{tools} (\beta A)^{\frac{5-\alpha}{4n+\alpha+1}} E^{\frac{8}{4n+\alpha+1}} (kd)^{-\frac{3+\alpha}{4n+\alpha+1}} (Q)^{-\frac{4m-2\alpha+2}{4n+\alpha+1}}$$

(21)

Now, the exponent on erosion rate varies between 0.27 and 1.05. As before, slope-area scaling cannot be evaluated in a meaningful manner, since the dependence of $\beta$ and $d$ on area is unknown.

In the cover-dominated case, eq. (7) reduces to

$$S_{cover} = \left(\frac{Q_s}{K_{bl}W}\right)^{1/n} Q^{-\frac{m}{n}} = \left(\frac{EQ_s}{K_{bl}^2 kd}\right)^{1/2n} Q^{-\frac{m}{n}}$$

(22)

Here, I also used the approximation $W \gg d$, and channel width was eliminated using eq. (4). For rivers crossing an uplifting fault block, where all parameters apart from erosion rate can be treated constant, slope scales with incision rate $E^{1/2n}$, with the exponent lying in the range of 0.07-0.33, using a range of $n$-values of 1.5-7, as discussed above. For catchments in a topographic steady state $Q_s$ can be expected to scale linearly with erosion rate (eq. 14), yielding a slope equation of the form

$$S_{cover} = \left(\frac{\beta A E^2}{K_{bl}^2 kd}\right)^{1/2n} Q^{-\frac{m}{n}}$$

(23)

In this case, the exponent on erosion rate yields the range of values of 0.14-0.67. The dependence on $Q_s$ introduces an additional dependence on area, affecting the slope-area exponent. Assuming that $Q$ is proportional to drainage area ($c = 1$), and $m = 1$ and $n = 2$, the slope-area exponent evaluates to -0.25. However, both bedload fraction and sideward deflection distance can be expected to scale with drainage area in an unknown way, which would alter the relationship. In addition, if $E$ varies systematically along the stream, the slope-area scaling will be affected. For example, if $E$ decreases in the downstream direction, it also decreases with increasing drainage area, resulting in an increase of the concavity index. This is in line with observations.

In summary, the values for the scaling exponents for the relationship between slope and erosion rates for the different cases that have been discussed encompass the range of observed values (Table 1). All four observations regarding channel bed slope, as outlined in the beginning of this chapter, can be obtained.

### 3.4 Sinuosity

Recapitulating the results of Stark et al. (2010), we expect sinuosity to increase both with increasing erodibility $k$ and increasing storm strike frequency. After substituting eq. (4) into eq. (9) to eliminate channel width and employing the approximation $W \gg d$, the tools-dominated case gives

$$\sigma_{tools} = \left(\frac{K_{bl}M_0 Ukd}{Q_s E}\right)^{1/n} S_V Q^{\frac{m}{n}}$$

(24)

As before, the bedload particle speed $U$ is dependent on slope and discharge. Accounting for this gives

$$\sigma_{tools} = \frac{S_V}{k_{tools}}\left(\frac{kd}{E}\right)^{\frac{3+\alpha}{4n+\alpha+1}} (Q_s)^{\frac{\alpha-5}{4n+\alpha+1}} (Q - Q_c)^{\frac{4m-2\alpha+2}{4n+\alpha+1}}$$

(25)

Here, I have also replaced discharge $Q$ with effective discharge $Q - Q_c$, subtracting a critical discharge for the onset of bedload motion $Q_c$ (e.g., Montgomery and Buffington, 1997; Rickenmann, 2001), which is important when considering discharge variability (e.g., Lague et al., 2005; Molnar, 2001), and thus sinuosity dependence on storm frequency. In the cover-dominated case, we get

$$\sigma_{cover} = \left(\frac{K_{bl}{}^2 k d}{E Q_s}\right)^{1/2n} S_V (Q - Q_c)^{\frac{m}{n}}$$

(26)

For the following discussion, $S_V$ is treated as a constant, but could in principle be a function of local tectonics and, therefore, implicitly erosion rate. The expected scaling with erodibility is directly obvious from both eqs. (25) and (26); sinuosity scales with $k^{(3+\alpha)/(4n+\alpha+1)}$ in the tools-dominated case, and with $k^{1/2n}$ in the cover-dominated case. Since there is currently no accepted

way of measuring $k$, no quantitative data exist and the comparison cannot go further.

Next, we link sinuosity to the variability of precipitation. The variability of forcing parameters is important for threshold processes (e.g., Lague, 2010), and the only relevant threshold process that we have considered is bedload transport. When considering variable forcing, mean discharge needs to be replaced by the effective discharge $Q_{eff}$ that determines bedload

transport and incision on long time scales (e.g., Lague et al., 2005; Molnar, 2001). In general, if the threshold discharge is higher than the mean discharge, a higher discharge variability results in a higher effective discharge (Deal, 2017). In storm-driven catchments, such as the streams on the Pacific Arc islands studied by Stark et al. (2010), geomorphically active floods are generally rare (e.g., Molnar, 2001) and erosion is limited to a few days per year and often less, making this assumption valid. Variability in discharge $V_Q$ scales with frequency of large storms $F_{Storm}$ (cf. Deal, 2017; Rossi et al., 2016). We thus find

a scaling that agrees with the observations of Stark et al. (2010):

$$\sigma \sim Q_{eff} \sim V_Q \sim F_{Storm}$$

(27)

**4 Discussion**

**4.1 Comparison to previous models**

The model proposed here connects channel width, bed slope, and sinuosity to discharge, erosion rate, and substrate erodibility,

via the core variable of bed cover. It fills a gap within the available published models, as it is a 1D reach-scale model constructed from considerations of the physics of bedload transport and fluvial erosion, without the need of arbitrary closing assumptions. I have used a fluvial bedrock erosion model (eq. 3) that includes both tools and cover effects, and that is consistent

with current process understanding (e.g., Beer et al., 2017; Fuller et al., 2016; Johnson and Whipple, 2010; Sklar and Dietrich, 2004), as well as quantitative field and laboratory measurements (Auel et al., 2017; Beer and Turowski, 2015; Chatanantavet and Parker, 2009; Inoue et al., 2014; Johnson and Whipple, 2010). The model presented here thus improves upon existing 1D reach-scale models both in the plausibility of the underlying assumptions, and, as has been shown in section 3, in the predictive

power concerning the observed scaling relationships. In addition, the model is complete in the sense that it does not feature a lumped calibration parameter with obscure physical meaning. All model parameters have a direct physical interpretation and can, at least in principle, be measured in the laboratory or the field.

## 4.2 Sideward deflection of bedload

To further validate or refine the model, we need information on some of the unconstrained parameters. In particular, we are missing observations on bedload paths in partially alluviated beds and on sideward deflection of bedload particles. While no data is available on the former, at least some initial observations have been reported on the latter. From laboratory observations, Fuller et al. (2016) argued that roughness dominantly controls sideward deflection of bedload in bedrock channels and therefore lateral erosion. This interpretation is supported by field data of Beer et al. (2017). For a full quantification of the model, the

sideward deflection length scale $d$ would need to be measured for a realistic range of boundary conditions, varying hydraulics, bed roughness, particle size and characteristics. To upscale the model to the reach scale, we would need scaling relationships of bed roughness with drainage area or other morphological parameters that vary along a stream. A comprehensive investigation of the controls on the scaling of bed roughness of bedrock channels is not known to the author. An additional complication arises from the role of alluvium. An alluviated bed is typically rougher than bedrock (e.g., Chatanantavet and

Parker, 2008; Ferguson et al., 2017; Hodge et al., 2011; 2016), and the effect of stationary sediment on a bedrock bed on sideward deflection of moving particles has not yet been investigated.

We can obtain some tentative constraints on these scaling relationships by considering catchments in a topographic steady state. I assume that, in the cover-dominated domain, sideward deflection length scale $d$ and bedload fraction $\beta$ are dependent

on drainage area $A$ according to a power law, with exponents $a$ and $b$, respectively. The slope-area scaling can be written as

$$S_{cover} \sim \left(\frac{\beta}{d}\right)^{1/2n} A^{\frac{1}{2n} - c\frac{m}{n}} \sim A^{\frac{b-a}{2n}} A^{\frac{1}{2n} - c\frac{m}{n}} = A^{\frac{b-a+1}{2n} - c\frac{m}{n}}$$

(28)

Here, I used the hydraulic scaling (eq. 17) to replace discharge with area. If we assume that the concavity index, which includes both the explicit and implicit dependence on drainage area in eq. (28), is equal to 1/2, and use $m = 1$, $n = 2$ and $c = 1$ (see

section 3.2), we obtain $b-a = 1$. Similarly, assuming that the width-area scaling in eq. (15) should have an exponent of 1/2, from the width equation (15), we obtain

$$W \sim (\beta d)^{1/2} A^{1/2} \sim A^{\frac{a+b}{2}} A^{1/2} = A^{\frac{a+b+1}{2}}$$

(29)

This yields $a+b = 0$. Solving, we obtain $a = 1/2$ and $b = -1/2$. This means that the sideward deflection length $d$ increases when moving downstream while the bedload fraction $\beta$ decreases, both with the square root of drainage area. At least for the bedload fraction, this seems to be a plausible value (see Turowski et al., 2010). For $d$, at first glance, an increase with drainage area seems somewhat surprising, since it is often assumed that roughness decreases in the downstream direction (e.g., Ferguson, 2007; Nitsche et al., 2012). However, this assumption is made for alluvial channels and is related to downstream fining that is observed in many alluvial streams (e.g., Parker, 1991). In a bedrock channel, it seems plausible that a progressive increase in cover leads to an overall increase in roughness when moving downstream.

## 4.3 Implications for stream-profile inversion

The theoretical framework of the stream power model has been frequently used to obtain information about tectonic uplift or fluvial erosion rates by stream-profile inversion (e.g., Kirby and Whipple, 2001; Wobus et al., 2006b). Within the stream power framework, the steady state profile of bedrock channels is given by

$$S = \left(\frac{E}{k_e}\right)^{1/n'} A^{-c\frac{m'}{n'}}$$

(30)

Here, $k_e$ is a lumped calibration parameter that is commonly interpreted to reflect bedrock erodibility. For the analysis, it is usually assumed that $m' = 0.5$, $n' = 1$ and $c = 1$ (see Lague, 2014), to obtain a concavity index equal to 1/2, although evidence points to $n'$ typically being larger than one (DiBiase and Whipple, 2011; Harel et al., 2016; Lague, 2014). Then, slope is fitted with a power law against area and a value for $E/k_e$ can be derived. More sophisticated inversions exploit the transient dynamics of models that can resolve erosion histories and find separate fit solutions for both $E$ and $k_e$ (e.g., Roberts and White, 2010). Comparing eq. (30) to the four slope equations obtained by the model (eqs. 20-23), the steady state equations show the same power-law dependence of slope $S$ on drainage area $A$ and erosion rate $E$, although, depending on the domain (cover- vs tools-dominated) and the type of forcing (crossing a fault or topographic steady state), the scaling exponents differ. In particular, the relationship between the scaling exponent of slope with erosion rate and the scaling exponent on drainage area (the concavity index) may be different to the one inferred from eq. 30. For example, for steady state catchments in the cover-dominated domain, the scaling exponent on erosion rate evaluates to $1/n$ (eq. 23), while the concavity index evaluates to $(1 + c\,m)/n$ (eq. 28; using $b - a = 1$). Further, the physical interpretation of $m$ and $n$ is different from the interpretation of $m'$ and $n'$. While in the stream power model, $m'$ and $n'$ are directly related to the mechanics of fluvial bedrock erosion, in the model proposed here, $m$ and $n$ are related to the mechanics of bedload transport. Clearly, a wrong choice for the value of $n'$ in particular leads to incorrect estimates of erosion rates. If $m'$ and $n'$ are determined by bedload transport, as suggested here, $n'$ may fall in the plausible range between 1.5 and 7 (see section 3.2), and could be very different from $n' = 1$ that is typically used when deriving tectonic information from stream-profile inversion.

**4.4 The role of cover for sinuous bedrock channels**

Here, I have argued that in streams where impact erosion is the dominant fluvial erosion process, cover is the central variable that needs to be considered. Nevertheless, it can be expected that bed cover modulates sinuosity development also in streams where other erosion processes are dominant. As has been argued by Johnson and Finnegan (2015), the dominant erosion process – slaking or impact erosion – determines whether a particular stream actively meanders or not in their study region. However, even weak rock that can be worn away by clear water flow will not erode if it is covered by a thick layer of sediment. Moreover, arguably, wetting-drying cycles are both less frequent and less efficient when water needs to flow through the pores of a gravel or sand layer. Although the erosion mechanism may likely make certain channels more prone to active meandering than others, I suggest here that bed cover plays a role in all of them.

**5 Conclusion**

Based on the idea that relative sediment supply controls bedrock channel meandering (Moore, 1926; Shepherd, 1972), and by making links to lateral erosion and channel width evolution, a physics-based 1D model of bedrock channel morphology was constructed. The model correctly predicts the observed scaling relations between channel width and slope with discharge and erosion rate, and sinuosity with erodibility and storm strike frequency. In addition, it yields plausible ranges of values of the exponent values and can explain why a channel should develop to a steady state sinuosity. The model is rooted in process physics, is fully parameterised and does not include lumped calibration parameters. It therefore describes bedrock channel morphology more completely than previously proposed models.

By predicting steady state long-profiles of bedrock channels similar to the stream power model, the model proposed here explains the success of the stream power model in describing steady state channel bed slope and its failure to account for the scaling of width. In addition, it reconnects channel long-profile analysis with the insights that have been obtained on the physics of fluvial bedrock erosion over the last two decades. If the physical argument proposed here is correct, methods of stream profile inversion to obtain data on erosion rate or tectonic history using the stream power model are based on incorrect assumptions. The results obtained with these methods are likely incorrect, especially if they were used to derive uplift histories. A further interesting point is that, if the model is correct, the scaling of width with erosion rate (eq. 4) seems less complicated than that of slope (eq. 7), since its dependence on the details of the hydraulics and hydrology is less pronounced. This may indicate that tectonic information can be more robustly obtained from channel width than from slope.

The model proposed here opens a new view to reach-scale bedrock channel morphology. Although the assumptions that have been made are physically plausible, many of them are as yet untested and little data are available to constrain the values of and the controls on some of the key parameters, such as the sideward deflection length scale. Nevertheless, the strong rooting of

the model in process physics and its success in correctly predicting scaling relationships of slope, width and sinuosity is encouraging and warrants further investigation. For a comprehensive evaluation of the model and the underlying assumptions, we need detailed investigations of the sediment dynamics in partially alluviated bedrock channels. In particular, this includes bedload transport equations for particles moving over a bare bedrock bed, maps of bedload particle concentrations in the channel for various bed morphologies and flow conditions, and research into the controls on sideward deflection of moving particles.

## Appendix A

In the tools-dominated domain, the channel bed slope is given by the equation (eq. 19)

$$S_{tools} = \left(\frac{Q_s{}^2}{M_0 U K_{bl} W^2}\right)^{1/n} Q^{-\frac{m}{n}}$$

(A1)

Here, the bedload particle speed U depends on shear stress and therefore slope and discharge. Based on laboratory flume measurements, Auel et al. (2017) gave an equation (their equation 19) for particle speed as a function of shear stress, including various previous measurements, both over bedrock and alluvial beds

$$U = 1.46 \left(\frac{1}{\rho}\left(\frac{\tau}{\tau_c} - 1\right)\right)^{1/2}$$

(A2)

To eliminate Shields stress, I use the DuBoys equation and the water continuity equation

$$\tau = \rho g H S$$

(A3)

$$Q = WHV$$

(A4)

Water flow velocity $V$ can be computed by the variable power flow resistance equation, which can be expressed as a function of slope, discharge and width (Ferguson, 2007; Nitsche et al., 2012)

$$V = k_V (gS)^{\frac{1-\alpha}{2}} R^{\frac{1-3\alpha}{2}} \left(\frac{Q}{W}\right)^{\alpha}$$

(A5)

Here, $R$ is a measure of bed roughness with dimensions of length, for example the standard deviation of the bed surface (e.g., Nitsche et al., 2012), and $\alpha \approx 0.6$ is a constant. Shear stress can then be written as

$$\tau = \frac{\rho}{k_V} (gS)^{\frac{\alpha+1}{2}} R^{\frac{3\alpha-1}{2}} \left(\frac{Q}{W}\right)^{1-\alpha}$$

(A6)

For substitution into A1, I neglect the threshold (i.e., $\tau/\tau_c$-1 ≈ $\tau/\tau_c$) to obtain

$$U = \frac{1.46}{\sqrt{\tau_c k_V}} (gS)^{\frac{\alpha+1}{4}} R^{\frac{3\alpha-1}{4}} \left(\frac{Q}{W}\right)^{\frac{1-\alpha}{2}}$$

(A7)

$$S_{tools} = k_{tools} (Q_s)^{\frac{8}{4n+\alpha+1}} (W)^{-\frac{6+2\alpha}{4n+\alpha+1}} (Q)^{-\frac{4m-2\alpha+2}{4n+\alpha+1}}$$

(A8)

Here, $k_{tools}$ is assumed to be constant

$$k_{tools} = (g)^{-\frac{\alpha+1}{4n+\alpha+1}} R^{\frac{1-3\alpha}{4n+\alpha+1}} \left(\frac{\sqrt{\tau_c k_V}}{1.46 M_0 K_{bl}}\right)^{\frac{4}{4n+\alpha+1}}$$

(A9)

Substituting the width equation (eq. 4)

$$S_{tools} = k_{tools} \left(\frac{E}{kd}\right)^{\frac{3+\alpha}{4n+\alpha+1}} (Q_s)^{\frac{5-\alpha}{4n+\alpha+1}} (Q)^{-\frac{4m-2\alpha+2}{4n+\alpha+1}}$$

(A10)

**Notation**

| | | |
|---|---|---|
| | $A$ | Drainage area [m$^2$]. |
| | $A_{cover}$ | Covered bed area [m$^2$]. |
| 5 | $A_{tot}$ | Total bed area [m$^2$]. |
| | $a$ | Scaling exponent, $d$-$A$. |
| | $b$ | Scaling exponent, $\beta$-$A$. |
| | $C$ | Fraction of covered bed. |
| | $C_c$ | Critical cover. |
| 10 | $c$ | Scaling exponent, $Q$-$A$. |
| | $d$ | Sideward deflection length scale, reach [m]. |
| | $d_x$ | Sideward deflection length scale, cross section [m]. |
| | $d_{xy}$ | Sideward deflection length scale, at a point [m]. |
| | $E$ | Vertical erosion rate [m/s]. |
| 15 | $e$ | Base of the natural logarithm. |
| | $F_{Storm}$ | Storm strike frequency [s$^{-1}$]. |
| | $g$ | Acceleration due to gravity [m/s$^2$]. |
| | $H$ | Water depth [m]. |
| | $K_{bl}$ | Bedload transport efficiency [kg m$^{-3m}$s$^{-m}$]. |
| 20 | $k$ | Erodibility [m$^2$/kg]. |
| | $k_e$ | Erodibility in stream power model [m$^{1-3m}$s$^{1-m}$]. |
| | $k_h$ | Hydrology coefficient [m$^{3-2c}$/s]. |
| | $k_s$ | Steepness index [m$^{2\theta}$]. |
| | $k_{tools}$ | Lumped constant, tools-dominated channel slope. |
| 25 | $k_V$ | Velocity coefficient [m$^{2\alpha}$]. |
| | $L$ | Reach length [m]. |
| | $L_V$ | Straight length from reach start to end [m]. |
| | $M_0$ | Minimum mass per area necessary to cover the bed [kg/m$^2$]. |
| | $m$ | Discharge exponent in bedload equation. |
| 30 | $m'$ | Discharge exponent in the stream power model. |
| | $n$ | Slope exponent in bedload equation. |
| | $n'$ | Slope exponent in the stream power model. |
| | $Q$ | Water discharge [m$^3$/s]. |
| | $Q_c$ | Critical discharge for the onset of bedload motion [m$^3$/s]. |

| | | |
|---|---|---|
| $Q_c^*$ | | Relative sediment supply at the critical cover. |
| $Q_{eff}$ | | Effective discharge [m³/s]. |
| $Q_s$ | | Upstream sediment mass supply [kg/s]. |
| $Q_s^*$ | | Relative sediment supply; sediment transport rate over transport capacity. |
| 5 | $Q_t$ | Mass sediment transport capacity [kg/s]. |
| | $R$ | Bed roughness length scale [m]. |
| | $S$ | Channel bed slope. |
| | $S_{cover}$ | Channel bed slope predicted in the cover-dominated approximation. |
| | $S_{tools}$ | Channel bed slope predicted in the tools-dominated approximation. |
| 10 | $S_V$ | Valley slope. |
| | $U$ | Bedload speed [m/s]. |
| | $V$ | Water flow velocity [m/s]. |
| | $V_Q$ | Discharge variability parameter. |
| | $W$ | Channel width [m]. |
| 15 | $W_{cover}$ | Covered length within the channel width [m]. |
| | $\alpha$ | Scaling exponent, $V$-$Q$. |
| | $\beta$ | Fraction of sediment transported as bedload. |
| | $\theta$ | Concavity index; scaling exponent $S$-$A$. |
| | $\rho$ | Density of water [kg/m³]. |
| 20 | $\sigma$ | Sinuosity. |
| | $\sigma_{cover}$ | Sinuosity predicted in the cover-dominated approximation. |
| | $\sigma_{tools}$ | Sinuosity predicted in the tools-dominated approximation. |
| | $\tau$ | Bed shear stress [N/m²]. |
| | $\tau_c$ | Critical bed shear stress at the onset of bedload motion [N/m²]. |
| 25 | | |

**Competing interests**

The author declares that he has no conflict of interest.

**Acknowledgements**

I thank J. Scheingross and E. Deal for insightful discussions. The model presented here is rooted in discussions with C. Stark and J. Barbour about a decade ago. The author was prompted to take up the problem of bedrock channel sinuosity again during a field visit to the Chinese Tien Shan in 2016, studying rivers affected by active folding with K. Cook. M. D'Arcy, J.
Scheingross and A. Bufe as well as two anonymous reviewers looked through a previous version of this paper and their helpful comments led to tremendous improvements.

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

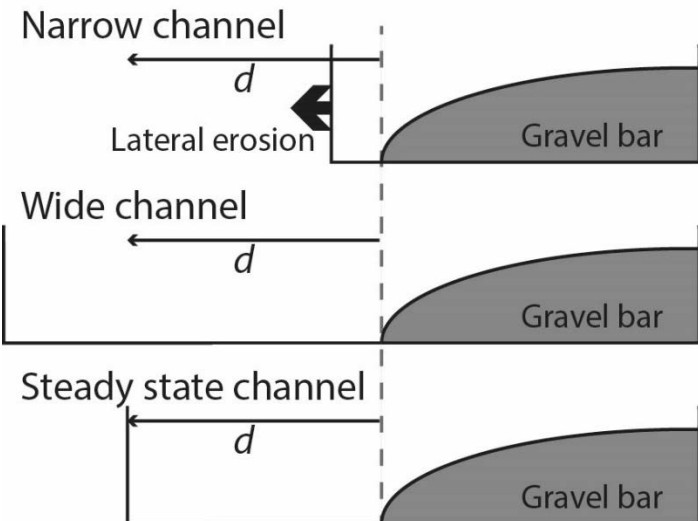

**Figure 1: Illustration of how the sideward deflection length scale _d_ and the channel width interact to determine lateral erosion. The dashed vertical line shows the relevant deflection point within the cross section. Top: in a narrow channel, particles that are laterally deflected a distance _d_ may hit the wall and cause erosion. The channel widens. Center: in a wide channel, the deflected particles do not reach the wall. No lateral erosion occurs. Conversely, few particles travel over the bedrock bed near to the wall. Sufficient tools to drive the vertical erosion of the bed are only available within the distance _d_. An inner channel with the steady state width is formed. Bottom: in a steady state channel, the channel walls are just out of reach of the deflected particles.**

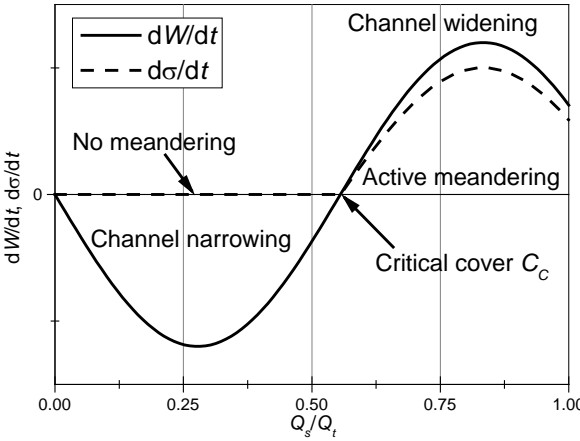

**Figure 2: Schematic relation between the rate of change of width _dW/dt_ (black line) and sinuosity _dσ/dt_ (dashed line) with relative sediment supply $Q_s/Q_t$. At low supply, no sediment particles impact the walls, the channel narrows, and does not meander actively. At high supply, frequent sediment impacts on the channel walls drive lateral erosion, leading to channel widening and active**

meandering. At the critical cover, the rate of change of width is zero. The exact position of this point depends on absolute channel width.

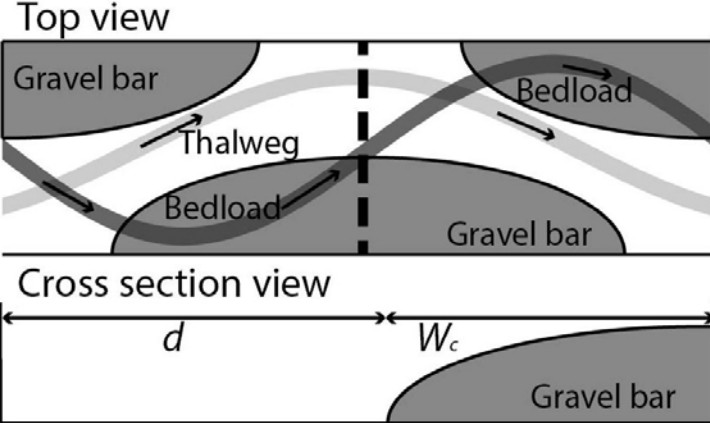

Figure 3: Top: Schematic drawing of the top view of the channel with alternating gravel bars (dark grey), thalweg and main water pathway (light grey), and main bedload path way (transparent dark grey) after Bunte et al. (2006). Uncovered bedrock is depicted in white. Bottom: Cross section across the centre of a bar (dotted black line in the top view), where the bedload path crosses from the bar into the uncovered channel. This cross section is relevant for setting the reach-scale channel width, since the sideward deflection of bedload particles toward the left-hand wall should be maximised (cf. Fig. 4). At steady state, the uncovered width within
the cross section should be equal to the sideward deflection length scale $d$, and the relation $d + W_c = W$ should hold (cf. Fig. 1; eq. 2).

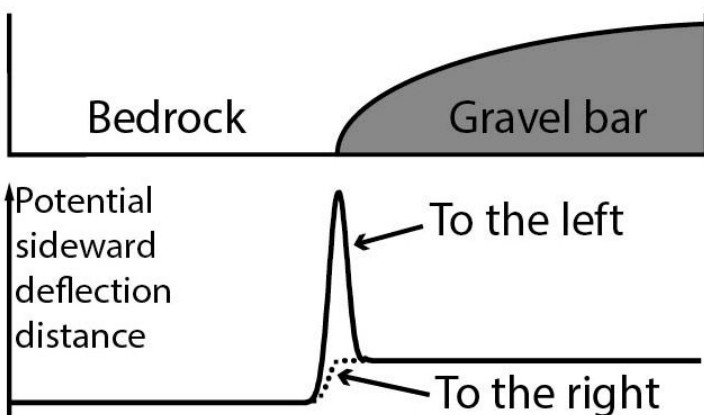

Figure 4: The potential sideward deflection distance is larger over alluvium than bedrock, since roughness elements facilitate sideward deflection of moving particles. However, the same roughness elements block path of the deflected particles, thus limiting
the total distance. The largest deflection distances occur at the boundary between alluvium and bedrock towards the bedrock bed. Only where the particle stream intersects this point can large sideward deflection distances be achieved.

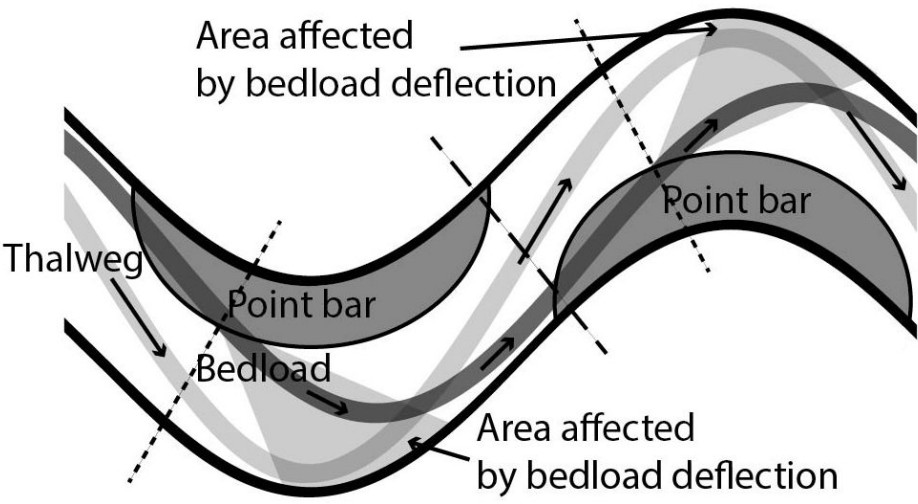

**Figure 5: Schematic illustration of the thalweg (light grey) and gravel bedload path (dark grey) through a meandering channel, after the observations of Dietrich and Smith (1984) and Julien and Anthony (2002). Flow is from left to right. Dotted lines show the relevant cross section for particle deflection. Areas that are presumably affected by bedload particle deflection and should lead to wall erosion are shaded in light grey. The dashed line is placed at the inflection point of the channel centre line.**

Table 1: Data sets and scaling exponents used for model evaluation, as reported by Lague (2014).

| | | River / Region (observations) Domain (predictions) | Scaling exponents | | Reference |
|---|---|---|---|---|---|
| | | | Width-erosion rate | Slope-erosion rate | |
| Channels crossing a fault | Observations | Bakeya | -0.63 | 0.49 | Lavé and Avouac, 2001 |
| | | Peikang | -0.42 | 0 | Yanites et al., 2010 |
| | Model prediction | Tools-dominated | -0.5 | 0.12-0.47 | |
| | | Cover-dominated | -0.5 | 0.07-0.33 | |
| Steady state catchments | Observations | Eastern Tibet | N.A. | 0.65 | Ouimet et al., 2009 |
| | | San Gabriel Mountains | 0 | 0.49 | DiBiase et al., 2010 |
| | | Mendocino Triple Junction | 0 | 0.25 | Snyder et al., 2000 |
| | | Siwalik hills | N.A. | 0.93 | Kirby and Whipple, 2001; Wobus et al., 2006b |
| | Model prediction | Tools-dominated | 0 | 0.27-1.05 | |
| | | Cover-dominated | 0 | 0.14-0.67 | |