# Peer review of "Alluvial cover controlling the width, slope and sinuosity of bedrock channels"

_Earth Surface Dynamics, 2017_

## Referee Comment (RC1) · Anonymous Referee #1 · 8 Nov 2017

Overall summary:

'Alluvial cover controlling the width, slope and sinuosity of bedrock channels' by J.M. Turowski proposes a new, process-physics based, model for the erosion, morphology, and scaling of bedrock river channels at steady-state conditions. It is a well-written, detailed, and innovative piece of research that fits strongly within the realms of Earth Surface Dynamics and I believe would make an excellent contribution to the fluvial geomorphology community. The manuscript is the first, to my knowledge, to develop a model that considers the physics of the processes that generate meandering and sinusoity in bedrock, and the manuscript essentially has two parts: model development (section 2) and model testing against field observations (section 3). A comprehensive review (Lague, 2014) set out the fundamental requirements for any successful bedrock

river modelling approach and the author here tests his new model against these requirements, as well as two additional relations specifically for channel sinuosity observed in the field. The new model performs well, with the scaling exponents of the new model for slope and width matching the relatively broad range found in field data (Table 1). The model also finds a scaling of sinuosity that agrees with observations of discharge variability and erodibility in previous work.

In the discussion, the advantages of the new modelling approach are highlighted against existing approaches, and some implications of the new method for one of key applications of bedrock river erosion modelling; stream-profile inversion to determine the history of tectonic uplift or fluvial erosion rates. These aspects of the manuscript are important, demonstrating the implications of the new model for the fluvial geomorphology community rather than simply presenting the new model.

While I do not have any problems with the proposed model itself, and definitely think the work is appropriate for publication in Earth-Surface Dynamics, there are several relatively minor details related to the presentation of the work that I believe the author should address in order to help the reader engage with the work and to maximise the potential impact. These are discussed below, followed by some minor detailed comments (e.g. typographical errors).

Detailed points:

I feel that the manuscript would benefit from some additional figures and visual representation of the work, particularly in the introductory and model development sections. The text of the manuscript does a good job in explaining the proposed framework of the model, but I think it would help the reader if this was also represented in a conceptual figure in order to visualise the main aims of the proposed modelling approach, the definition of the important parameters (i.e. d).

For example, a top view of a channel showing an idealised bedrock meander showing the development of sinuosity through the path of bedload detaching from flow lines to

**ESurfD**
impact the outer bank would make a nice simple addition to the Introduction (Section 1). Immediately the reader would understand the potential importance of bedload sediment transport in driving sinuosity, and help give further justification for developing the new model with bedrock cover as the key parameter (Page 4, Line 6). Another example of a potential new figure could explain the conditions required for steady-state width to develop (in section 2). On Page 5, Line 9, the text even states 'for the purposes of illustration', but there is no visual illustration. There could, for example, be one panel of a figure where $d < W$, and the channel is actively widening, one where $d \sim W$ and one where $d > W$, and bedload impacts have reduced and channel widening ceased. Such figures would help the reader visualise the background to the proposed modelling approach, and complement the description of the processes currently explained purely in the text (and equations).

The other main point is associated with the structure of the manuscript; in particular, the location of the section where the new modelling approach is compared to existing approaches (Section 4.1). Section 4.1 does a good job of identifying the gaps and weaknesses associated with existing approaches, and how there remains a need to develop and use an approach that is purely physics-based, without assumptions. I think some of these elements could be moved to the introduction, as additional justification for the development of a new model. The current introduction mainly focuses on the sinuosity problem, yet this is just one of three parts of the model (alongside more general scaling of width and slope). A suggestion to the author could be to take elements from the initial part of the discussion and integrate it into the introduction, providing a more comprehensive discussion of the bedrock modelling problem that the new approach goes on to tackle.

Developing from this last point, due to the current structure of the paper, it feels like section 4.3 is tagged on at the end of the manuscript. This section is really important and interesting, and potentially of wide interest to readers beyond the immediate field of bedrock erosion modelling, towards wider landscape evolution applications. If the

**ESurfD**
paper introduces the current issues with existing modelling approaches (current section 4.1) earlier, it could also discuss the potential issues associated with the selection of the scaling exponents m and n for applications of bedrock modelling in landscape evolution studies (i.e. stream profile inversion). This could then potentially give section 4.3 in the discussion more impact, as the reader is already aware of the importance of need to accurately constrain the values of m and n, and the discussion can highlight the differences between those commonly used in landscape evolution studies and the values identified using the new modelling approach. This is just a suggestion to potentially help develop the implications of the new modelling approach for wider landscape evolution problems.

Specific points (typographical errors etc).

Page 5, Line 10: Insert 'to' between 'Due' and 'frequent'.

P6, L22: 'river' should be 'rivers'

Figure 2: Typo of 'section' in bottom panel label.

P13, L9: 'adjust' should be 'adjusts'

P23, L17: space should be inserted to 'Ferguson,2007'

P25, L14: 'Storm strike frequency'

---

## Referee Comment (RC2) · Anonymous Referee #2 · 22 Nov 2017

The paper "Alluvial cover controlling the width, slope and sinuosity of bedrock channels" presents a model that incorporates sediment-flux driven bedrock erosion, and scaling of bedrock river channels' physical features to describe sinuosity at a steady state value and predict the observed relations between sinuosity, erodibility and storm frequency. The paper, the model, and the supportive arguments were well-constructed and clearly explained, making this an interesting and enjoyable paper to read. The paper references previous work in a manner that allows the reader a clear understanding of the basis for the model. Further, the novelty of the model presented is based on a solid foundation of previous work and sound logic. The methodology and assumptions are clearly outlined. Further, to my knowledge, the model presented is completely novel. I believe this will be a substantial contribute to the journal, and fits well within

the journal's scope, and the field at large.

Overall, I would rate the scientific significance and scientific quality of the paper as excellent. However, the presentation quality would benefit from additional graphical depictions of the model, and possibly the scaling data also. While the author has done a nice job of clearly taking the reader through the calculations of the model, I believe readers' understanding of the model and relationships described could be improved from additional depictions.

Additionally, I have the following minor notes on the rest of the text: • The "Tools-dominated" vs "Cover-dominated" could use a little more initial introduction to full appreciate the meaning and differences. The author discusses this a bit just after Table 1. However, it is difficult to relate how these equations differ relative to reality. While the author does describe the typical environments these two types of equations would apply to at the end of section 3, why these tools apply here could use more development. • The conclusions are concisely written; however, they may benefit from further development. I felt that additional development of the last paragraph of the paper in particular could benefit from additional development. While the paper does layout the novelty of the work, as the paper currently stands it doesn't sell the novelty and usefulness of what's been produced as well as it could. • I would reconsider the title or placement of section 4.4. The title doesn't seem to express what the author is saying. This paragraph could also be adjusted to be part of the conclusion. As it currently reads it seems a slightly out of place. • A minor error includes a few of the variables are undefined this the text (for example, Qt). Additionally, some of the variables (Scover, Stools, Ccover, Ctools, $\sigma$cover, and $\sigma$tools) are not listed in the notation list. Finally, the notation list is also slightly out of order.

---

## Author Comment (AC1) · 29 Nov 2017

Dear reviewers, dear editor, dear interested readers,

Thanks to all who have read the manuscript and provided comments. Here, I want to briefly address the main review comments. A detailed rebuttal letter will be supplied with the revised paper.

Both reviewers requested additional figures to illustrate the model concepts. I do agree that this will be helpful for the reader and have prepared three additional figures (see also attached pdf).

First, a figure to illustrate the relationship between channel width and the sideward deflection length scale, both in transient adjustment and in steady state (Fig. 1).

Second, a figure illustrating the potential for sideward deflection at various points in the cross section (Fig. 2).

Third, a figure illustrating the thalweg and gravel path in a meandering channel (Fig. 3).

Please also note the supplement to this comment:
https://www.earth-surf-dynam-discuss.net/esurf-2017-46/esurf-2017-46-AC1-supplement.pdf

Narrow channel

*d*

Lateral erosion

Gravel bar

Wide channel

*d*

Gravel bar

Steady state channel

*d*

Gravel bar

**Fig. 1.** Interaction of sideward deflection length and channel width.

![Figure 2 showing a cross section with Bedrock and Gravel bar, and potential sideward deflection distance with "To the left" and "To the right" curves]

**Fig. 2.** Potential sideward deflection distance in a cross section.

Area affected
by bedload deflection

Thalweg

Point bar

Point bar

Bedload

Area affected
by bedload deflection

**Fig. 3.** Bedload path through a sinuous channel.

[Figure]

**Supplement:**

Dear reviewers, dear editor, dear interested readers,

Thanks to all who have read the manuscript and provided comments. Here, I want to briefly address the main review comments. A detailed rebuttal letter will be supplied with the revised paper.

Both reviewers requested additional figures to illustrate the model concepts. I do agree that this will be helpful for the reader and have prepared three additional figures.

First, a figure to illustrate the relationship between channel width and the sideward deflection length scale, both in transient adjustment and in steady state (figure numbers as in the revised manuscript).

[Figure]

**Figure 1: Illustration of how the sideward deflection length scale *d* and the channel width interact to determine lateral erosion. The dashed vertical line shows the relevant deflection point within the cross section. Top: in a narrow channel, particles that are laterally deflected a distance *d* may hit the wall and cause erosion. The channel widens. Center: in a wide channel, the deflected particles do not reach the wall. No lateral erosion occurs. Bottom: in a steady state channel, the channel walls are just out of reach of the deflected particles.**

Second, a figure illustrating the potential for sideward deflection at various points in the cross section.

[Figure]

**Figure 4: The potential sideward deflection distance is larger over alluvium than bedrock, since roughness elements facilitate sideward deflection of moving particles. However, the same roughness elements block path of the deflected particles, thus limiting the total distance. The largest deflection distances occur at the boundary between alluvium and bedrock towards the bedrock bed. Only where the particle stream intersects this point can large sideward deflection distances be achieved.**

Third, a figure illustrating the thalweg and gravel path in a meandering channel.

[Figure]

**Figure 5: Schematic illustration of the thalweg (light grey) and gravel bedload path (dark grey) through a meandering channel, after the observations of Dietrich and Smith (1984) and Julien and Anthony (2002). Flow is from left to right. Dotted lines show the relevant cross section for particle deflection. Areas that are presumably affected by bedload particle deflection and should lead to wall erosion are shaded in light grey. The dashed line is placed at the inflection point of the channel centre line.**

Although none of the reviewers has picked up on this, there was a definitional ambiguity of the sideward deflection length scale. The symbol *d* was used for the deflection distance at a particular point, the distance relevant for lateral erosion within a specific cross section, and for the value relevant for setting width at the reach scale. I have now clarified these different parameters and will introduce them with separate symbols in the revised paper.

Both reviewers made some comments on structure. In the revised paper, the section on previous models (in the discussion, section 4.1) will be moved to the opening of the model development section (section 2). In addition, I will add a paragraph to the introduction introducing the wider relevance of bedrock channels, for example for deriving tectonic information using stream profile inversion. I will also slightly expand the conclusion.

---

## Author Comment (AC2) · 16 Dec 2017

I thank the two reviewers for taking the time to read and comment on my manuscript. Their criticisms and comments are very appreciated and in trying to address everything, I hope I have improved the paper sufficiently to warrant publication.

Below, after I make some general statements and replies to comments that were made by both reviewers, the reviewers' comments are reproduced in normal font, while my replies are given in *Italics*.

Additional illustrations

Both reviewers requested additional figures to illustrate the model concepts. I do agree that this will be helpful for the reader and have added three additional figures.

First, a figure to illustrate the relationship between channel width and the sideward deflection length scale, both in transient adjustment and in steady state (figure numbers as in the revised manuscript).

[Figure]

**Figure 1: Illustration of how the sideward deflection length scale *d* and the channel width interact to determine lateral erosion. The dashed vertical line shows the relevant deflection point within the cross section. Top: in a narrow channel, particles that are laterally deflected a distance *d* may hit the wall and cause erosion. The channel widens. Center: in a wide channel, the deflected particles do not reach the wall. No lateral erosion occurs. Conversely, few particles travel over the bedrock bed near to the wall. Sufficient tools to drive the vertical erosion of the bed are only available within the distance *d* of the deflection point. An inner channel with the steady state width is formed. Bottom: in a steady state channel, the channel walls are just out of reach of the deflected particles.**

Second, a figure illustrating the potential for sideward deflection at various points in the cross section.

[Figure]

Figure 4: The potential sideward deflection distance is larger over alluvium than bedrock, since roughness elements facilitate sideward deflection of moving particles. However, the same roughness elements block path of the deflected particles, thus limiting the total distance. The largest deflection distances occur at the boundary between alluvium and bedrock towards the bedrock bed. Only where the particle stream intersects this point can large sideward deflection distances be achieved.

Third, a figure illustrating the thalweg and gravel path in a meandering channel.

[Figure]

Figure 5: Schematic illustration of the thalweg (light grey) and gravel bedload path (dark grey) through a meandering channel, after the observations of Dietrich and Smith (1984) and Julien and Anthony (2002). Flow is from left to right. Dotted lines show the relevant cross section for particle deflection. Areas that are presumably affected by bedload particle deflection and should lead to wall erosion are shaded in light grey. The dashed line is placed at the inflection point of the channel centre line.

Although none of the reviewers has picked up on this, there was a definitional ambiguity of the sideward deflection length scale. The symbol d was used for the deflection distance at a particular point, the distance relevant for lateral erosion within a specific cross section, and for

the value relevant for setting width at the reach scale. I have now clarified these different parameters and introduced separate symbols.

Finally, I have gone through the entire text and edited to improve clarity and readability.

Reviewer comments

Reviewer #1

'Alluvial cover controlling the width, slope and sinuosity of bedrock channels' by J.M. Turowski proposes a new, process-physics based, model for the erosion, morphology, and scaling of bedrock river channels at steady-state conditions. It is a well-written, detailed, and innovative piece of research that fits strongly within the realms of Earth Surface Dynamics and I believe would make an excellent contribution to the fluvial geomorphology community. The manuscript is the first, to my knowledge, to develop a model that considers the physics of the processes that generate meandering and sinuosity in bedrock, and the manuscript essentially has two parts: model development (section 2) and model testing against field observations (section 3). A comprehensive review (Lague, 2014) set out the fundamental requirements for any successful bedrock river modelling approach and the author here tests his new model against these requirements, as well as two additional relations specifically for channel sinuosity observed in the field. The new model performs well, with the scaling exponents of the new model for slope and width matching the relatively broad range found in field data (Table 1). The model also finds a scaling of sinuosity that agrees with observations of discharge variability and erodibility in previous work.
In the discussion, the advantages of the new modelling approach are highlighted against existing approaches, and some implications of the new method for one of key applications of bedrock river erosion modelling; stream-profile inversion to determine the history of tectonic uplift or fluvial erosion rates. These aspects of the manuscript are important, demonstrating the implications of the new model for the fluvial geomorphology community rather than simply presenting the new model.
While I do not have any problems with the proposed model itself, and definitely think the work is appropriate for publication in Earth-Surface Dynamics, there are several relatively minor details related to the presentation of the work that I believe the author should address in order to help the reader engage with the work and to maximise the potential impact. These are discussed below, followed by some minor detailed comments (e.g. typographical errors).
*Thank you for the kind assessment and the detailed comments.*

Detailed points:
I feel that the manuscript would benefit from some additional figures and visual representation of the work, particularly in the introductory and model development sections. The text of the manuscript does a good job in explaining the proposed framework of the model, but I think it would help the reader if this was also represented in a conceptual figure in order to visualise the main aims of the proposed modelling approach, the definition of the important parameters (i.e. d).
*I have added several new figures and illustrations (see general replies).*

For example, a top view of a channel showing an idealised bedrock meander showing the development of sinuosity through the path of bedload detaching from flow lines to impact the outer bank would make a nice simple addition to the Introduction (Section 1). Immediately the reader would understand the potential importance of bedload sediment transport in driving

sinuosity, and help give further justification for developing the new model with bedrock cover as the key parameter (Page 4, Line 6).
*I have decided not to provide such a figure in the introduction, but a similar figure is now in section 2.3, where the model is extended to sinuous channels (new figure 5).*

Another example of a potential new figure could explain the conditions required for steady-state width to develop (in section 2). On Page 5, Line 9, the text even states 'for the purposes of illustration', but there is no visual illustration. There could, for example, be one panel of a figure where d < W, and the channel is actively widening, one where d _ W and one where d > W, and bedload impacts have reduced and channel widening ceased. Such figures would help the reader visualise the background to the proposed modelling approach, and complement the description of the processes currently explained purely in the text (and equations).
*Such a figure was added to section 2.1 (new figure 1).*

The other main point is associated with the structure of the manuscript; in particular, the location of the section where the new modelling approach is compared to existing approaches (Section 4.1). Section 4.1 does a good job of identifying the gaps and weaknesses associated with existing approaches, and how there remains a need to develop and use an approach that is purely physics-based, without assumptions. I think some of these elements could be moved to the introduction, as additional justification for the development of a new model. The current introduction mainly focuses on the sinuosity problem, yet this is just one of three parts of the model (alongside more general scaling of width and slope). A suggestion to the author could be to take elements from the initial part of the discussion and integrate it into the introduction, providing a more comprehensive discussion of the bedrock modelling problem that the new approach goes on to tackle.
*I have moved the overview over previous models to the start of section 2, rather than the introduction. It feels more naturally placed at this point.*

Developing from this last point, due to the current structure of the paper, it feels like section 4.3 is tagged on at the end of the manuscript. This section is really important and interesting, and potentially of wide interest to readers beyond the immediate field of bedrock erosion modelling, towards wider landscape evolution applications. If the paper introduces the current issues with existing modelling approaches (current section 4.1) earlier, it could also discuss the potential issues associated with the selection of the scaling exponents m and n for applications of bedrock modelling in landscape evolution studies (i.e. stream profile inversion). This could then potentially give section 4.3 in the discussion more impact, as the reader is already aware of the importance of need to accurately constrain the values of m and n, and the discussion can highlight the differences between those commonly used in landscape evolution studies and the values identified using the new modelling approach. This is just a suggestion to potentially help develop the implications of the new modelling approach for wider landscape evolution problems.
*I have added a paragraph at the start of the paper bringing up the wider relevance of bedrock channels and the issue of stream-profile inversion.*

Specific points (typographical errors etc).
Page 5, Line 10: Insert 'to' between 'Due' and 'frequent'.
P6, L22: 'river' should be 'rivers'
Figure 2: Typo of 'section' in bottom panel label.
P13, L9: 'adjust' should be 'adjusts'
P23, L17: space should be inserted to 'Ferguson,2007'
P25, L14: 'Storm strike frequency'
*I have corrected all typos.*

Reviewer #2

The paper "Alluvial cover controlling the width, slope and sinuosity of bedrock channels" presents a model that incorporates sediment-flux driven bedrock erosion, and scaling of bedrock river channels' physical features to describe sinuosity at a steady state value and predict the observed relations between sinuosity, erodibility and storm frequency. The paper, the model, and the supportive arguments were well-constructed and clearly explained, making this an interesting and enjoyable paper to read. The paper references previous work in a manner that allows the reader a clear understanding of the basis for the model. Further, the novelty of the model presented is based on a solid foundation of previous work and sound logic. The methodology and assumptions are clearly outlined. Further, to my knowledge, the model presented is completely novel. I believe this will be a substantial contribute to the journal, and fits well within the journal's scope, and the field at large.

Overall, I would rate the scientific significance and scientific quality of the paper as excellent. However, the presentation quality would benefit from additional graphical depictions of the model, and possibly the scaling data also. While the author has done a nice job of clearly taking the reader through the calculations of the model, I believe readers' understanding of the model and relationships described could be improved from additional depictions.

*I thank the referee for the constructive comments. I have added several new graphics and illustrations (see general comments). The new graphics illustrate model assumptions. I have not added a figure with the validation data, as I do not see the added value. The data has been published, compiled and re-published in several contributions (compilations and general discussion in, e.g., Turowski et al., 2009, Yanites and Tucker, 2010, Lague, 2014). Lague (2014) in particular discussed the data in detail, and gave all the necessary graphics and fit statistics. Since within my paper, the only relevant value is the scaling exponent (summarised in Table 1), a plot of the data would not add anything that is of relevance.*

Additionally, I have the following minor notes on the rest of the text:

The "Tools-dominated" vs "Cover-dominated" could use a little more initial introduction to full appreciate the meaning and differences. The author discusses this a bit just after Table 1. However, it is difficult to relate how these equations differ relative to reality. While the author does describe the typical environments these two types of equations would apply to at the end of section 3, why these tools apply here could use more development.

*I have added an explanation for both tools- and cover-dominated domains. The sentences now read:*

First, in the tools-dominated domain, cover is scarce and bedrock erosion rate is controlled by the availability of tools.

Second, in the cover-dominated domain, tools are abundant, but most of the bed is covered. Then, the erosion rate is set by the fraction of the exposed bedrock.

The conclusions are concisely written; however, they may benefit from further development. I felt that additional development of the last paragraph of the paper in particular could benefit from additional development. While the paper does layout the novelty of the work, as the paper currently stands it doesn't sell the novelty and usefulness of what's been produced as well as it could.

*I have added a few sentences detailing the needs for further study. I do see the point of the reviewer that the conclusions could be further expanded, but I do not want to merely repeat the points made in the discussion (for sake of brevity and mental sanity of the reader). With >14000 words, the paper is long already. Currently, I cannot think of any more points that need to be mentioned.*

I would reconsider the title or placement of section 4.4. The title doesn't seem to express what the author is saying. This paragraph could also be adjusted to be part of the conclusion. As it currently reads it seems a slightly out of place.
*Changed to 'The role of cover for sinuous bedrock channels'.*

A minor error includes a few of the variables are undefined this the text (for example, Qt). Additionally, some of the variables (Scover, Stools, Ccover, Ctools, _cover, and _tools) are not listed in the notation list.
Finally, the notation list is also slightly out of order.
*I have updated and corrected the notation list.*

---

## Author Response (AR2)

Dear editor,

Please find my final manuscript attached. I did some cosmetic changes to the text, fixing some typos and generally improving the language. I have also added a sentence to the conclusion, making the point that channel width seems less complicated than channel slope and may therefore give more robust results when inverting channel morphology for tectonic information.

Thanks for your work and help.

All the best and happy 2018, Jens Turowski

[revised manuscript text omitted]